# Statistical Collusion by Collectives on Learning Platforms

Etienne Gauthier [1]   Francis Bach [1]   Michael I. Jordan [1 2]

## Abstract

As platforms increasingly rely on learning algorithms, collectives may form and seek ways to influence these platforms to align with their own interests. This can be achieved by coordinated submission of altered data. To evaluate the potential impact of such behavior, it is essential to understand the computations that collectives must perform to impact platforms in this way. In particular, collectives need to make a priori assessments of the effect of the collective before taking action, as they may face potential risks when modifying their data. Moreover they need to develop implementable coordination algorithms based on quantities that can be inferred from observed data. We develop a framework that provides a theoretical and algorithmic treatment of these issues and present experimental results in a product evaluation domain.

## 1. Introduction

The dynamic interaction among agents and algorithms creates a complex ecosystem where unanticipated individual and collective behavior can emerge. The study of such behavior is crucial for understanding how to design systems that are robust, fair, and aligned with societal values.

In a network, agents often have diverse motivations and multiple incentives. When the incentives of the interacting agents do not fully align with those of the designer of the learning system, the former may wish to influence the learning process. This becomes salient in the common case in which a learning system interacts with a large number of agents. Such agents may collaborate, forming a cartel, by pooling their data and devising a common strategy. Accordingly, even if the learning algorithm is robust to single-agent

[1]Inria, Ecole Normale Supérieure, PSL Research University, Paris [2]Departments of EECS and Statistics, University of California, Berkeley. Correspondence to: Etienne Gauthier <etienne.gauthier@inria.fr>.

*Proceedings of the 42$^{nd}$ International Conference on Machine Learning*, Vancouver, Canada. PMLR 267, 2025. Copyright 2025 by the author(s).

adversarial behavior, the collective may be able to exert a significant influence on the algorithm. This concept of collective action has its origins in economic theories (Olson, 1965) and has been more recently explored in the context of machine learning (Hardt et al., 2023). It is crucial to understand how such collectives can influence learning algorithms to better comprehend the dynamics within the network of agents and to deploy algorithms that are reliable and aligned with the majority interests of consumers. We will focus specifically on the case of a platform that deploys a learning algorithm and interacts with a population of consumers.

**Collective Action in Machine Learning.** We draw on the work of Hardt et al. (2023), who investigate the following problem: a population of individuals $(x, y) \in \mathcal{X} \times \mathcal{Y}$, drawn i.i.d. from a distribution $\mathcal{D}$, interacts with a platform. A collective of relative size $\alpha \in (0, 1)$ forms within this population with the goal of influencing the platform. The collective's influence is quantified by a success metric $S(\alpha)$, the definition of which depends on the collective's objective. To influence the platform, members of the collective modify their data according to a common strategy $h : \mathcal{X} \times \mathcal{Y} \to \mathcal{X} \times \mathcal{Y}$, which maps the original features and labels of the data to modified features and labels. The platform observes a mixture of distributions $\alpha\tilde{\mathcal{D}} + (1-\alpha)\mathcal{D}$, where $\tilde{\mathcal{D}}$ is the distribution of $h(z)$ with $z \sim \mathcal{D}$. The platform selects a classifier based on this distribution. Thus, the objective for the collective is to choose a strategy $h$ that maximizes its success $S(\alpha)$. Hardt et al. (2023) propose strategies $h$ for two distinct goals: *signal planting* and *signal erasing*. They derive lower bounds on the success $S(\alpha)$, enabling the identification of a minimum collective size $\alpha^*$ such that $S(\alpha) \geq S^*$ for all $\alpha \geq \alpha^*$, where $S^*$ is a target threshold.

Unfortunately, several of the strategies $h$ studied by Hardt et al. (2023) are not available in practice to the collective. Furthermore, the bounds obtained on the success $S(\alpha)$ depend on key parameters that are unknown to the collective. Lastly, in practice, the collective seeks a priori guarantees of success before altering its data, as modifying the data may expose members to risk. In this paper, we introduce a new framework to enable members of the collective to learn strategies $h$ efficiently and to infer the parameters that determine their success on the platform.

**Contributions.** Our primary contribution is the introduction of a novel framework which empowers collectives via statistical inference. This statistical framework allows for the derivation of three key results, each tied to a distinct objective that a collective may pursue to influence a platform: signal planting, signal erasing, and a new objective we introduce, *signal unplanting*.

We thoroughly explore strategies that a collective can employ to achieve each of these goals and provide theoretical guarantees for the effectiveness of these strategies. The statistical inference capabilities of the collective serve two main purposes: first, to estimate the most effective strategies for influencing the platform, and second, to infer key parameters that determine the collective's success. This dual approach allows the collective to predict its potential impact on the platform with high probability.

Our framework reduces to that of Hardt et al. (2023) in the infinite data regime, although we note that we improve on earlier results in this regime—our lower bounds are tighter than those from Hardt et al. (2023). Our main focus, however, is the setting in which the signal set is finite. We obtain bounds in this setting that form staircase-like curves. This result provides a new interpretation of data poisoning bounds: the success of attacks depends not just on the signal set to be poisoned as a whole but more precisely on each feature within it. Each feature has a resistance to poisoning, requiring a specific level of attack to breach it, resulting in these staircase patterns. When the steps are close enough, these curves resemble the smooth sigmoid shapes seen in data poisoning literature.

We construct a synthetic dataset to validate our theoretical findings and to examine the influence of various parameters empirically. Our empirical results highlight, among other things, that the effectiveness of the collective depends not only on its *relative* size compared to the total number of agents interacting with the platform but also on its *absolute* size. A larger collective, in absolute terms, can obtain better statistical estimates and thus more accurately infer optimal strategies. This result implies that larger platforms may be more vulnerable to collective action.

## 2. Related Work

Our research builds upon and extends the concept of collective action in machine learning, a framework originally introduced by Hardt et al. (2023). Collective action relates closely to data poisoning attacks in machine learning, a subset of security attacks that disrupt model training by injecting malicious data to degrade performance or alter predictions. Of particular virulence, backdoor attacks embed a hidden trigger in the data that activates malicious behavior only when the trigger appears, making them subtle and hard to detect. For comprehensive discussions on data poisoning, backdoor attacks, and defense mechanisms, we refer the reader to the surveys by Tian et al. (2022), Guo et al. (2021), and Cinà et al. (2023).

Data poisoning is a critical topic in machine learning. Many empirical studies focus on backdoor attacks and the defense mechanisms for learning algorithms. However, there is relatively little research that analyzes the effectiveness of these attacks theoretically. Grosse et al. (2022) show that backdoor patterns induce a stable representation of the target class. The classifier relies on the backdoor trigger and disregards other features. In the case of binary classification, Manoj & Blum (2021) demonstrate that if the model has a property called nonzero memorization capacity, then a successful backdoor attack is possible. The model's vulnerability is assessed based on its ability to memorize out-of-distribution values. In particular, overparameterized linear models have higher memorization capacity and are more susceptible to attacks. Xian et al. (2023) also investigate the context of binary classification and propose a hypothesis regarding the distribution of poisoned data, which allows them to derive useful results on the effectiveness of an attack. Wang et al. (2024) explore the effectiveness of backdoor attacks from a statistical standpoint. They provide bounds on the statistical risks associated with a poisoned model, specifically analyzing how these risks manifest when the model is evaluated on both clean and backdoored data for a finite sample size. Li & Liu (2024) present a theoretical examination of a backdoor attack applied to a convolutional neural network with two layers. Cinà et al. (2024) conduct an empirical study on the learning curves associated with backdoor attacks. Moreover, they demonstrated that classifiers with stronger regularization are generally more resistant to poisoning attacks, although this comes with a slight decrease in accuracy on clean data.

What sets the concept of collective action apart conceptually is its treatment of the collective as a group of individuals, each representing a data point. This perspective also has economic, social, and political dimensions, as certain groups of individuals can unite and collaborate to influence decisions. Such ideas have been explored in areas of research at the intersection of machine learning and other fields (Vincent et al., 2019; Albert et al., 2020; Vincent et al., 2021; Albert et al., 2021; Creager & Zemel, 2021; Vincent & Hecht, 2021). See Appendix A of Hardt et al. (2023) for an in-depth analysis of the related work on collective action. In addition, Ben-Dov et al. (2024) highlight how the learning algorithm shapes the success of collective action.

An important contribution of Hardt et al. (2023) is to study data poisoning via the formalism of Bayes-optimal classification, which yields a conceptual inversion of the idea of strategic classification (Hardt et al., 2016). While strategic

classification revolves around a firm's ability to anticipate and respond to the actions of a single, strategic individual, collective action shifts the focus toward a scenario where individuals collectively anticipate and strategically respond to the optimizing behavior of the firm. This concept has also been explored by Zrnic et al. (2021). Unlike traditional strategic classification, which primarily considers the firm's perspective, collective action highlights the role of workers and consumers on online platforms.

# 3. Statistical Algorithmic Collective Action in Classification

First, we will describe the new setting for deriving theoretical bounds in collective action that are effectively computable by the collective. We then present our three main results, which address three different objectives for the collective. Two of these objectives are classic goals of collective action: signal planting and signal erasing. Additionally, we introduce a new objective: signal unplanting, where the collective aims to prevent an association between the signal set and a certain label. We will explore strategies and provide theoretical guarantees for each of these objectives.

## 3.1. Setting

We consider a platform that deploys a learning algorithm in a universe $\mathcal{X} \times \mathcal{Y}$. We assume that $\mathcal{X} \times \mathcal{Y}$ is finite. Each individual corresponds to a single data point $(x, y) \in \mathcal{X} \times \mathcal{Y}$.

The platform trains a classifier $\hat{f}$ on a training dataset. The training dataset is composed of $N$ consumers which are initially drawn i.i.d. according to some distribution $\mathcal{D}$. Among these consumers, a certain number $n < N$ forms a collective to strategically influence the firm's behavior. The collective shares a common strategy $h : \mathcal{X} \times \mathcal{Y} \to \mathcal{X} \times \mathcal{Y}$. The $N - n$ base consumers and the $n$ members of the collective together form an empirical distribution of consumers $\hat{\mathcal{P}}$, and the corresponding dataset constitutes the training set. The collective wants to obtain guarantees on the influence they have on the platform at test time.

**Notation.** Given a distribution $\mathcal{Q}$ over $\mathcal{X} \times \mathcal{Y}$, we denote by $\mathcal{Q}_{\mathcal{X}}$ the marginal distribution over features. We will simply write $\mathcal{Q}$ when the context allows. We denote by $\tilde{\mathcal{D}}$ the distribution of $h(z), z \sim \mathcal{D}$. More generally, for a dataset $D$, we write $\tilde{D} := \{h(z) \mid z \in D\}$ (as a multiset) for the same dataset after applying strategy $h$. We will also write, for $E \subseteq \mathcal{X} \times \mathcal{Y}$, $\hat{\mathbb{P}}_{z \sim D}(z \in E) := \frac{1}{\#D} \sum_{z_i \in D} \mathbb{1}_{\{z_i \in E\}}$ the empirical probability of the event $E$ induced by a dataset $D$. We use $\hat{\mathbb{P}}$ to denote empirical probabilities, and $\mathbb{P}$ for population probabilities. When the variables do not need to be explicitly stated, we may write $\hat{\mathbb{P}}_D(E)$ and $\mathbb{P}_D(E)$ instead of $\hat{\mathbb{P}}_{z \sim D}(z \in E)$ and $\mathbb{P}_{z \sim D}(z \in E)$ respectively.

**The collective.** Given a test set of consumers $D_{\text{test}} \overset{i.i.d.}{\sim} \mathcal{D}$, the collective's goal is to obtain guarantees with high probability on their success $\hat{S}(n)$ as a function of $D_{\text{test}}$. The definition of $\hat{S}(n)$ is based on the objective desired by the collective. We will consider three objectives: signal planting, signal unplanting, and signal erasing. The collective modifies its data using strategy $h$ to maximize its success.

We assume that the collective has access to the value $N$. This is a weak assumption because it is common in practice. For example, if the platform is a polling institute seeking to understand participants' voting preferences based on their demographic data, the total number of people surveyed is usually publicly available.

Unless stated otherwise, the collective has access only to their own data and not to the data of consumers who are not part of the collective. The collective can pool its data to infer quantities that depend on the underlying distribution. Throughout this paper, we will use Hoeffding's concentration inequality (Lemma D.1) for simplicity. We will denote Hoeffding error terms as follows:

$$R_\delta(k) := \sqrt{\frac{\log(1/\delta)}{2k}},$$

for any $\delta > 0$ and $k \in \mathbb{N}^*$. We note that Hoeffding's inequality can be loose, for example when applied to sums of Bernoulli random variables with means close to zero. One can address this issue by using other concentration inequalities such as Bernstein; the framework we present here can readily incorporate such choices.

**The platform.** The firm observes an empirical distribution of consumers $\hat{\mathcal{P}}$. It selects a classifier $\hat{f}$ based on this distribution $\hat{\mathcal{P}}$. Following Hardt et al. (2023), we characterize classifiers $\hat{f}$ by their suboptimality in terms of total variation distance with respect to the observed distribution $\hat{\mathcal{P}}$:

**Definition 3.1.** Let $\varepsilon > 0$. A classifier $\hat{f} : \mathcal{X} \to \mathcal{Y}$ is $\varepsilon$-suboptimal on a set $\mathcal{X}' \subseteq \mathcal{X}$ under the distribution $\hat{\mathcal{P}}$ if there exists a distribution $\tilde{\tilde{\mathcal{P}}}$ with $TV(\hat{\mathcal{P}}, \tilde{\mathcal{P}}) \leq \varepsilon$ such that

$$\hat{f}(x) \in \underset{y \in \mathcal{Y}}{\operatorname{argmax}} \, \tilde{\mathcal{P}}(x, y)$$

for all $x \in \mathcal{X}'$. Here, $TV(\hat{\mathcal{P}}, \tilde{\mathcal{P}}) := \underset{E \subseteq \mathcal{X} \times \mathcal{Y}}{\sup} |\hat{\mathcal{P}}(E) - \tilde{\mathcal{P}}(E)|$ denotes the total variation distance between $\hat{\mathcal{P}}$ and $\tilde{\mathcal{P}}$.

The parameter $\varepsilon$ roughly controls how much the classifier can make use of statistics that go beyond simple frequency counts. It accounts for classifiers that consider feature interactions and capture complex patterns in the data.

## 3.2. Signal planting

In signal planting, we are given a transformation $g : \mathcal{X} \to \mathcal{X}$ and a target label $y^* \in \mathcal{Y}$. The map $g$ induces a signal set

defined by $\tilde{\mathcal{X}} := \{g(x) \mid x \in \mathcal{X}\}$. The success is defined as

$$\hat{S}(n) := \underset{x \sim D_{\text{test}}}{\hat{\mathbb{P}}} (\hat{f}(g(x)) = y^*),$$

where $D_{\text{test}} \overset{i.i.d.}{\sim} \mathcal{D}$ is the test set. In other words, the collective aims to enforce an association between the signal set $\tilde{\mathcal{X}}$ and a target label $y^*$ at test time.

### 3.2.1. Feature-label signal planting

A natural goal for the collective is to maximize its success, $\hat{S}(n)$. To ensure that its efforts are effective, the collective aims to establish theoretical lower bounds on $\hat{S}(n)$, as these provide a guarantee of success. To do so, the collective can play the feature-label signal planting strategy defined below.

**Definition 3.2** (Feature-label signal planting strategy)**.** We define the *feature-label signal planting strategy* as

$$h(x, y) = (g(x), y^*).$$

We analyze the effect of this strategy on the learning platform. Formally, we are given three independent datasets: a dataset $D^{(n)} \overset{i.i.d.}{\sim} \mathcal{D}$ of $n$ consumers which are part of the collective; a dataset $D^{(N-n)} \overset{i.i.d.}{\sim} \mathcal{D}$ of $N - n$ consumers which are not part of the collective; and a dataset $D_{\text{test}} \overset{i.i.d.}{\sim} \mathcal{D}$ of $N_{\text{test}}$ consumers forming the test dataset. We recall that $\tilde{D}^{(n)} := \{h(z) \mid z \in D^{(n)}\}$ (as a multiset). The training set of the platform's classifier is the concatenation of $\tilde{D}^{(n)}$ and $D^{(N-n)}$. In other words, the platform observes the distribution:

$$\hat{\mathcal{P}}(x_0, y_0) := \frac{n}{N} \underset{\tilde{D}^{(n)}}{\hat{\mathbb{P}}} (x_0, y_0) + \frac{N-n}{N} \underset{D^{(N-n)}}{\hat{\mathbb{P}}} (x_0, y_0).$$

The platform then chooses a classifier $\hat{f}$ based on $\hat{\mathcal{P}}$. We can now state the main result for signal planting:

**Theorem 3.3** (Signal planting lower bound, feature-label signal planting strategy)**.** *Let $\delta > 0$, and write $\tilde{\delta} := \delta/(2 + 2\#\tilde{\mathcal{X}} + 2\#\tilde{\mathcal{X}}\#\mathcal{Y})$. Then, by playing the feature-label signal planting strategy against a classifier that is $\varepsilon$-suboptimal on $\tilde{\mathcal{X}}$, the collective achieves with probability at least $1 - \delta$ (over the draw of the consumers):*

$$\begin{aligned}
\hat{S}(n) \geq \underset{\tilde{x} \sim \tilde{D}^{(n)}}{\hat{\mathbb{P}}} &\left[ \frac{n}{N} \left( \underset{\tilde{D}^{(n)}}{\hat{\mathbb{P}}} (\tilde{x}) - 2R_{\tilde{\delta}}(n) \right) \right. \\
&- \frac{N-n}{N} \left( \Delta_{\tilde{x}}^{(n)} + 2R_{\tilde{\delta}}(n) + 2R_{\tilde{\delta}}(N-n) \right) \quad (1) \\
&\left. - \frac{\varepsilon}{1-\varepsilon} > 0 \right] - R_{\tilde{\delta}}(n) - R_{\tilde{\delta}}(N_{\text{test}}),
\end{aligned}$$

*where* $\Delta_{\tilde{x}}^{(n)} := \underset{y' \in \mathcal{Y}\setminus\{y^*\}}{\max} \underset{D^{(n)}}{\hat{\mathbb{P}}} (\tilde{x}, y') - \underset{D^{(n)}}{\hat{\mathbb{P}}} (\tilde{x}, y^*).$

Proofs and additional remarks can be found in Appendix E. Note that lower bound (1) is fully computable by the collective as it depends only on datasets $D^{(n)}$ and $\tilde{D}^{(n)}$. $D^{(n)}$

is obtained through data pooling by the collective's members, and $\tilde{D}^{(n)}$ is computed by applying strategy $h$ to the dataset $D^{(n)}$. Details on the algorithm for computing the lower bound are provided in Appendix B, and the code is available at: `https://github.com/GauthierE/statistical-collusion`.

The interpretation of lower bound (1) is that success increases step by step: each feature $\tilde{x}$ in the signal set $\tilde{\mathcal{X}}$ has a certain *resistance* to being planted. As the relative size of the collective $n/N$ gradually increases, features $\tilde{x}$ are *cracked* as their resistance breaks, in decreasing order of resistance. For each feature $\tilde{x}$, its resistance depends on three terms. The first one, here $\frac{n}{N} \big( \underset{\tilde{D}^{(n)}}{\hat{\mathbb{P}}} (\tilde{x}) - 2R_{\tilde{\delta}}(n) \big)$, represents how prevalent the feature $\tilde{x}$ is in the modified data: the more frequently $\tilde{x}$ appears in the poisoned data, the greater the collective's ability to influence the associated label. The second term, here $-\frac{N-n}{N}(\Delta_{\tilde{x}}^{(n)} + 2R_{\tilde{\delta}}(n) + 2R_{\tilde{\delta}}(N-n))$, captures the counteracting influence of non-collective individuals in the population, quantifying how much they might limit the collective's success in planting the signal. It indicates how strongly the target label is associated with the signal set: the more frequent $y^*$ is in the signal set, the easier it is to plant the signal; if other labels are far more likely than $y^*$, planting the signal becomes more difficult. The third term, $-\frac{\varepsilon}{1-\varepsilon}$, represents the platform's ability to adapt to the distribution of its users. As $\varepsilon \mapsto \frac{\varepsilon}{1-\varepsilon}$ increases with $\varepsilon$, it benefits the collective to have $\varepsilon$ close to zero, limiting the platform's flexibility and resulting in a tighter bound.

The first term scales approximately linearly with $n$, by a factor of $n/N$, while the second term decreases by $1 - n/N$. However, the dependence is more complex than purely linear. The bound involves estimation terms $R_{\tilde{\delta}}(n)$, which decay at a rate proportional to $n^{-1/2}$ as $n$ increases. The bound also depends on $R_{\tilde{\delta}}(N-n)$, but these terms can be negligible as long as $n$ remains much smaller than $N$.

Also, the cardinality of $\tilde{\mathcal{X}}$ affects the definition of $\tilde{\delta}$, and thus the estimation terms $R_{\tilde{\delta}}$. The smaller $\#\tilde{\mathcal{X}}$ is, the better the collective's estimates will be, resulting in sharper bounds.

### 3.2.2. Feature-only signal planting

Note that the feature-label signal planting strategy assumes that the members of the collective can change both their features and their labels. This might not always be feasible in practice, where labels can be immutable. In this situation, a natural strategy for the collective is to change its feature $x$ to $g(x)$ when $y = y^*$. This strategy was explored by Hardt et al. (2023). However, it is preferable for the collective to additionally change its feature $x$ to some $x_0$ that is not in the signal set $\tilde{\mathcal{X}}$ when $y \neq y^*$. This ensures that not only does the feature belong to the signal set $\tilde{\mathcal{X}}$ when the label is $y^*$, but also that if a feature is in the signal set $\tilde{\mathcal{X}}$, the associated

label is necessarily $y^*$. This dual condition strengthens the association between the signal set $\tilde{\mathcal{X}}$ and the target label $y^*$. In this case, we can directly say that the term $\hat{\mathbb{P}}_{\tilde{D}^{(n)}}(g(x'), y')$ with $y' \neq y^*$ that would appear in the proof of Theorem 3.5 is equal to zero, leading to a sharper bound.

**Definition 3.4** (Feature-only signal planting strategy)**.** We define the feature-only signal planting strategy as

$$h(x, y) = \begin{cases} (g(x), y^*) & \text{if } y = y^*, \\ (x_0, y) & \text{otherwise,} \end{cases}$$

where $x_0 \in \mathcal{X} \backslash \tilde{\mathcal{X}}$ is any feature that does not belong to the signal set $\tilde{\mathcal{X}}$. Note that $x_0$ does not have to be fixed across all initial data points $(x, y)$.

We can generalize Theorem 3.3 to the feature-only strategy:

**Theorem 3.5** (Signal planting lower bound, feature-only signal planting strategy)**.** *Let $\delta > 0$, and write $\tilde{\delta} := \delta/(2 + 2\#\tilde{\mathcal{X}} + 2\#\tilde{\mathcal{X}}\#\mathcal{Y})$. Then, by playing the feature-only signal planting strategy against a classifier that is $\varepsilon$-suboptimal on $\tilde{\mathcal{X}}$, the collective achieves with probability at least $1 - \delta$ (over the draw of the consumers):*

$$\hat{S}(n) \geq \hat{\mathbb{P}}_{x' \sim D^{(n)}} \left[ \frac{n}{N} \left( \hat{\mathbb{P}}_{\tilde{D}^{(n)}}(g(x'), y^*) - 2R_{\tilde{\delta}}(n) \right) \right.$$
$$\left. - \frac{N-n}{N} \left( \Delta_{g(x')}^{(n)} + 2R_{\tilde{\delta}}(n) + 2R_{\tilde{\delta}}(N-n) \right) \right. \quad (2)$$
$$\left. - \frac{\varepsilon}{1-\varepsilon} > 0 \right] - R_{\tilde{\delta}}(n) - R_{\tilde{\delta}}(N_{\text{test}}),$$

*where $\Delta_{g(x')}^{(n)}$ is defined in Theorem 3.3.*

We can straightforwardly compare lower bounds (1) and (2). The only difference is that in (2), we have $\hat{\mathbb{P}}_{\tilde{D}^{(n)}}(g(x'), y^*)$ instead of $\hat{\mathbb{P}}_{\tilde{D}^{(n)}}(g(x'))$. This comparison directly measures the impact of not modifying labels on the collective's influence. In the following, we will always assume that the collective can modify its labels.

Signal planting relies on a straightforward strategy: flooding the platform with as many pairs $(g(x), y^*)$ as possible. This strategy is simple and does not require the collective to perform any statistical estimation. Statistical inference is only necessary for the collective to calculate the lower bound on success, not for defining the optimal strategy $h$. In the next two sections, we examine two objectives that additionally require statistical estimation to infer the optimal strategy $h$: signal unplanting and signal erasing.

**3.3. Signal unplanting**

The setting is essentially the same as before; the only difference is the success of the collective is defined as follows:

$$\tilde{S}(n) := \hat{\mathbb{P}}_{x \sim D_{\text{test}}} \left( \hat{f}(g(x)) \neq y^* \right).$$

The collective's goal is now to prevent an association between the signal set $\tilde{\mathcal{X}} = \{g(x) \mid x \in \mathcal{X}\}$ and the target label $y^*$.

3.3.1. NAIVE STRATEGY

A simple and naive strategy for the collective is to flood the platform with feature-label pairs of the form $(g(x), y')$ for some fixed $y' \neq y^*$. Indeed, for any $y' \neq y^*$, we have that

$$\hat{S}(n) = \hat{\mathbb{P}}_{x \sim D_{\text{test}}} \left( \hat{f}(g(x)) \neq y^* \right)$$
$$\geq \hat{\mathbb{P}}_{x \sim D_{\text{test}}} \left( \hat{f}(g(x)) = y' \right) =: \hat{S}_{y'}(n).$$

Therefore, the collective can compute lower bounds on $\hat{S}_{y'}(n)$ for all $y' \neq y^*$ using Theorem 3.3. It can then play the strategy $h(x, y) = (g(x), \bar{y})$ where $\bar{y}$ is the label that maximizes the lower bounds on $\hat{S}_{y'}(n)$ for $y' \neq y^*$. Note that this is equivalent to planting a signal with transformation $g$ and target label $\bar{y}$, so we obtain the same guarantees as in Theorem 3.3.

3.3.2. ADAPTIVE STRATEGY

The collective can also use an adaptive strategy, meaning that each member of the collective $(x, y)$ can change its feature-label pair to some $(g(x), y_{g(x)})$ where the modified label $y_{g(x)}$ depends on $g(x)$ and is no longer fixed. The plan for the collective is to estimate the optimal label using a subset of $n_e < n$ randomly chosen participants. Formally, we assume that $D^{(n)}$ is the concatenation of two independent datasets $D^{(n_e)}$ and $D^{(n-n_e)}$ drawn from $\mathcal{D}$. A natural strategy for the collective is to change every pair $(x, y)$ into $(g(x), \hat{y}_{g(x)})$ where for $\tilde{x} \in \tilde{\mathcal{X}}$:

$$\hat{y}_{\tilde{x}} := \underset{y' \in \mathcal{Y} \backslash \{y^*\}}{\arg\max} \hat{\mathbb{P}}_{D^{(n_e)}}(\tilde{x}, y'). \quad (3)$$

We formalize this strategy in the following definition.

**Definition 3.6** (Signal unplanting strategy)**.** We define the *signal unplanting strategy* as

$$h(x, y) = (g(x), \hat{y}_{g(x)}),$$

where $\hat{y}_{\tilde{x}}$ is defined in Equation (3).

Intuitively, this strategy means that the collective aims to select the most likely label among all labels different from $y^*$, given a feature $g(x)$.

**Theorem 3.7** (Signal unplanting lower bound)**.** *Let $\delta > 0$, and write $\tilde{\delta} := \delta/(2 + 6\#\tilde{\mathcal{X}})$. Let $n_e < n$. Then, by playing the signal unplanting strategy above against a classifier that is $\varepsilon$-suboptimal on $\tilde{\mathcal{X}}$, the collective achieves with*

*probability at least $1 - \delta$ (over the draw of the consumers):*

$$
\hat{S}(n) \geq \underset{\tilde{x} \sim \tilde{D}^{(n)}}{\hat{\mathbb{P}}} \left[ \frac{n}{N} \left( \underset{\tilde{D}^{(n)}}{\hat{\mathbb{P}}} (\tilde{x}) - 2R_{\tilde{\delta}}(n) \right) \right.
$$
$$
- \frac{N-n}{N} \left( \Delta_{\tilde{x}}^{(n-n_{\mathrm{e}})} + 2R_{\tilde{\delta}}(n - n_{\mathrm{e}}) + 2R_{\tilde{\delta}}(N - n) \right)
$$
$$
\left. - \frac{\varepsilon}{1 - \varepsilon} > 0 \right] - R_{\tilde{\delta}}(n) - R_{\tilde{\delta}}(N_{\mathrm{test}}),
$$

$$(4)$$

*where* $\Delta_{\tilde{x}}^{(n-n_{\mathrm{e}})} := \underset{D^{(n-n_{\mathrm{e}})}}{\hat{\mathbb{P}}} (\tilde{x}, y^*) - \underset{D^{(n-n_{\mathrm{e}})}}{\hat{\mathbb{P}}} (\tilde{x}, \hat{y}_{\tilde{x}}).$

### 3.4. Signal erasing

In the previous subsections, we examined how the collective can plant a signal by using a natural strategy, which consists in simply flooding the platform with feature-label pairs of the form $h(x, y) = (g(x), y^*)$, and how it can unplant a signal by estimating the most probable label different from $y^*$. In this section, we study another objective: signal erasing.

We consider a transformation $g : \mathcal{X} \to \mathcal{X}$. The success of the collective is now defined by:

$$
\hat{S}(n) := \underset{x \sim D_{\mathrm{test}}}{\hat{\mathbb{P}}} \left( \hat{f}(g(x)) = \hat{f}(x) \right).
$$

As outlined by Hardt et al. (2023), maximizing $\hat{S}(n)$ aligns with reducing the impact of $g$ on the learning algorithm. The term signal erasing is motivated by the example in tabular data where $g$ preserves certain features while removing others, for instance by setting some features to a fixed value. This effectively removes the impact of the erased features, provided these features are independent of the other ones.

In this part, we will make the following mild assumptions.

**Assumption (A1):** $\exists \eta > 0, \forall \tilde{x} \in \tilde{\mathcal{X}}, \exists y_{\tilde{x}}^* \in \mathcal{Y} : \forall y' \neq y_{\tilde{x}}^*, \underset{\mathcal{D}}{\mathbb{P}}(\tilde{x}, y_{\tilde{x}}^*) > \underset{\mathcal{D}}{\mathbb{P}}(\tilde{x}, y') + \eta.$

Intuitively, Assumption 1 implies two things. Firstly, each feature $\tilde{x} \in \tilde{\mathcal{X}}$ is sufficiently frequent in the base distribution. Secondly, given a feature $\tilde{x} \in \tilde{\mathcal{X}}$, there exists a label $y_{\tilde{x}}^*$ that is consequently more probable than the other labels in the base distribution.

**Assumption (A2):** *The transformation $g$ is idempotent:* $g(g(x)) = g(x)$ *for all $x \in \mathcal{X}$.*

To understand Assumption 2, consider data poisoning in image classification. Assumption 2 holds in data poisoning strategies where the trigger is a fixed, opaque watermark, as in the case of binary masks (Gu et al., 2019), where applying the mask twice is equivalent to applying it once. However, this is not true for strategies like pixel blending (Chen et al., 2017). In contrast, Assumption 2 naturally applies to tabular data, where the collective can apply a transformation $g$ to map a feature to a constant value.

Now, we outline a scheme that the collective can use to compute a lower bound. The basic idea is that each member $(x, y)$ of the collective keeps its feature $x$ but changes its label to the most likely label $y_{g(x)}^*$ based on the feature $g(x)$. This approach encourages the platform to predict the same label for both $x$ and $g(x)$, hence erasing the signal. The scheme is the following: first, the collective pools its data to predict the optimal $y_{\tilde{x}}^*$ for each $\tilde{x} \in \tilde{\mathcal{X}}$. Then, the collective applies some strategy $h$ based on the first step to erase the signal. Assuming the collective can compute the optimal label $y_{\tilde{x}}^*$ for each $\tilde{x} \in \tilde{\mathcal{X}}$, the erasure strategy is formally defined as follows:

**Definition 3.8** (Erasure strategy)**.** We define the erasure strategy as

$$
h(x, y) = (x, y_{g(x)}^*),
$$

where $y_{\tilde{x}}^*$ is given in Assumption 1 for each $\tilde{x} \in \tilde{\mathcal{X}}$.

We can now state the main result in signal erasing:

**Theorem 3.9** (Signal erasing lower bound)**.** *Let $\delta > 0$, and write $\tilde{\delta} := \delta/(2 + \#\tilde{\mathcal{X}}\#\mathcal{Y} + 2\#\mathcal{X} + 2\#\mathcal{X}\#\mathcal{Y})$. Assume that $\frac{2\log(1/\tilde{\delta})}{\eta^2} \leq n \leq N - \frac{2\log(1/\tilde{\delta})}{\eta^2}$ where $\eta$ is given in Assumption 1. Then, with probability at least $1 - \delta$ (over the draw of the consumers), the collective can compute $y_{\tilde{x}}^*$ for all $\tilde{x} \in \tilde{\mathcal{X}}$ and by playing the erasure strategy it achieves against a classifier that is $\varepsilon$-suboptimal on $\mathcal{X}$:*

$$
\hat{S}(n) \geq \underset{x' \sim D^{(n)}}{\hat{\mathbb{P}}} \left[ \frac{n}{N} \left( \underset{D^{(n)}}{\hat{\mathbb{P}}} (x') - 2R_{\tilde{\delta}}(n) \right) \right.
$$
$$
- \frac{N-n}{N} \left( \Delta_{x'}^{(n)} + 2R_{\tilde{\delta}}(n) + 2R_{\tilde{\delta}}(N - n) \right) \quad (5)
$$
$$
\left. - \frac{\varepsilon}{1 - \varepsilon} > 0 \right] - R_{\tilde{\delta}}(n) - R_{\tilde{\delta}}(N_{\mathrm{test}}),
$$

*where* $\Delta_{x'}^{(n)} := \underset{y' \in \mathcal{Y} \setminus \{y_{g(x')}^*\}}{\max} \underset{D^{(n)}}{\hat{\mathbb{P}}} (x', y') - \underset{D^{(n)}}{\hat{\mathbb{P}}} (x', y_{g(x')}^*).$

In signal erasing, just like in signal unplanting, the collective leverages its own data to compute the strategy $h$. However, the technique differs. In signal unplanting, the collective uses a fraction of its members to estimate the optimal label to play. Whereas in signal erasing, the collective, provided that it is sufficiently large, utilizes all of its data to compute the most likely label given a feature $\tilde{x}$, which exists under Assumption 1.

## 4. Experimental Evaluation

In our experiments, we simulate a platform that collects data on vehicles, where each sample represents a car. The features of each car include characteristics such as *Model Type*, *Fuel Type*, and *Country of Manufacture*. The labels assigned to each vehicle reflect the car's evaluation, categorized into four classes: Excellent, Good, Average, or Poor. Further

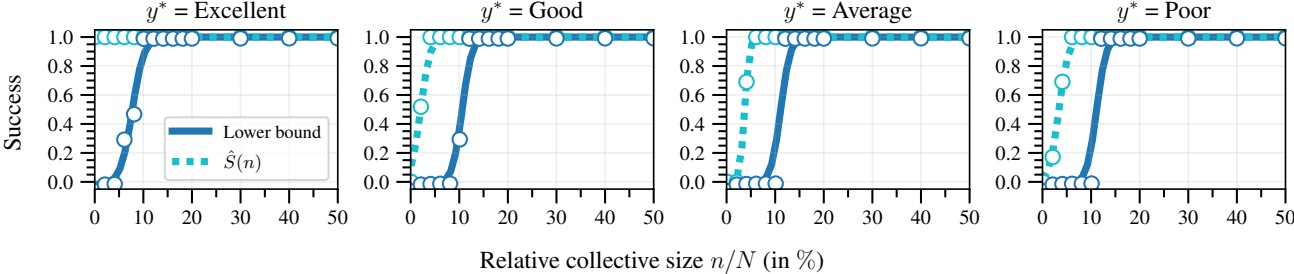

*Figure 1.* **Signal planting with feature-label strategy.** Comparison of the theoretical lower bound from Theorem 3.3 and the true success $\hat{S}(n)$ observed at test time for different values of $n$ and a fixed value of $N = 1,000,000$. For all target labels, the lower bound indicates that approximately 10% of the total number of agents interacting with the platform is necessary to significantly influence it. In reality, the success observed at test time shows that just under 5% of members is sufficient, except for the target label $y^*$ = Excellent, which is already consistently the most frequent and does not require any planting.

details on the dataset and the specific parameters used in the experiments can be found in Appendix A.

We consider a scenario where a collective seeks to influence the platform by lobbying against a particular category of vehicles, specifically SUVs with specific features. The collective defines a signal set $\tilde{\mathcal{X}}$ through a transformation $g$ fixing all feature values except *Country of Manufacture*. They may want to plant a signal targeting a label $y^*$ = Poor. They might also aim to unplant signals, specifically working to associate elements of $\tilde{\mathcal{X}}$ with labels $y \neq$ Excellent.

### 4.1. Signal planting

In Figure 1, we plot the lower bounds from Theorem 3.3 for various values of $n$ and compare them to the true success $\hat{S}(n)$ observed at test time. We fit sigmoid functions to interpolate the obtained values. The lower bounds are indeed lower than the success, and the gap between the two is not excessively large. This gap could potentially be narrowed using more advanced statistical inference methods. Interestingly, even though in practice the label $y^*$ = Excellent is already the most frequent label for every element of the signal set $\tilde{\mathcal{X}}$ and does not technically need to be planted—i.e., $\hat{S}(n) = 1$ for all $n$—the collective is still not guaranteed to have influence over the signal for small values of $n$. This stems from statistical uncertainties, which our framework highlights, causing the collective to consistently overestimate the number of agents needed for a given success.

Figure 2 shows how the lower bound evolves with the total number of individuals $N$. When the fraction $n/N$ is held fixed, increasing $N$ leads to a larger collective size $n$. This, in turn, improves the collective's ability to estimate key quantities, as reflected by the decreasing error terms $R_{\tilde{\delta}}(n)$. This suggests that platforms interacting with large user bases are more exposed to collectives altering their data. While this observation is specific to signal planting, it is even more relevant when optimal strategies need to be estimated, as in signal unplanting.

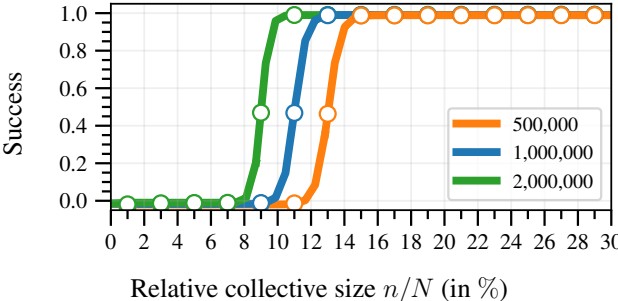

*Figure 2.* **Signal planting with feature-label strategy.** Lower bound from Theorem 3.3 with $y^*$ = Poor for different values of $n$ with $N = 500,000$, $N = 1,000,000$, and $N = 2,000,000$.

### 4.2. Signal unplanting

We now focus on signal unplanting with a target $y \neq$ Excellent, where the collective aims for the platform to predict a label other than Excellent for samples in the signal set $\tilde{\mathcal{X}}$.

The lower bound obtained in Theorem 3.7 depends on $n_e$, the size of the sub-collective used to determine the strategy $h$. As shown in Figure 3 (a), $n_e$ involves a trade-off: small values lead to erratic strategy estimates and weaker bounds, while overly large values increase the $R_{\tilde{\delta}}(n - n_e)$ term, also weakening the bound. A good balance is achieved at $n_e = 2,000$.

Figure 3 (b) compares this adaptive strategy with $n_e = 2,000$ to naive strategies planting labels $y^* \in$ {Good, Average, Poor}. The adaptive strategy consistently outperforms the naive ones by providing a higher lower bound, demonstrating the benefits of tailoring the strategy based on the data. However, it is worth noting that the naive strategy of planting the signal with $y^*$ = Good performs well for this dataset. This is due to the fact that most elements in the signal set $\tilde{\mathcal{X}}$ have Good as the second most likely label after Excellent.

In Figure 3 (c), we compare the lower bound with $n_e = 2,000$ and the actual success $\hat{S}(n)$ achieved at test time.

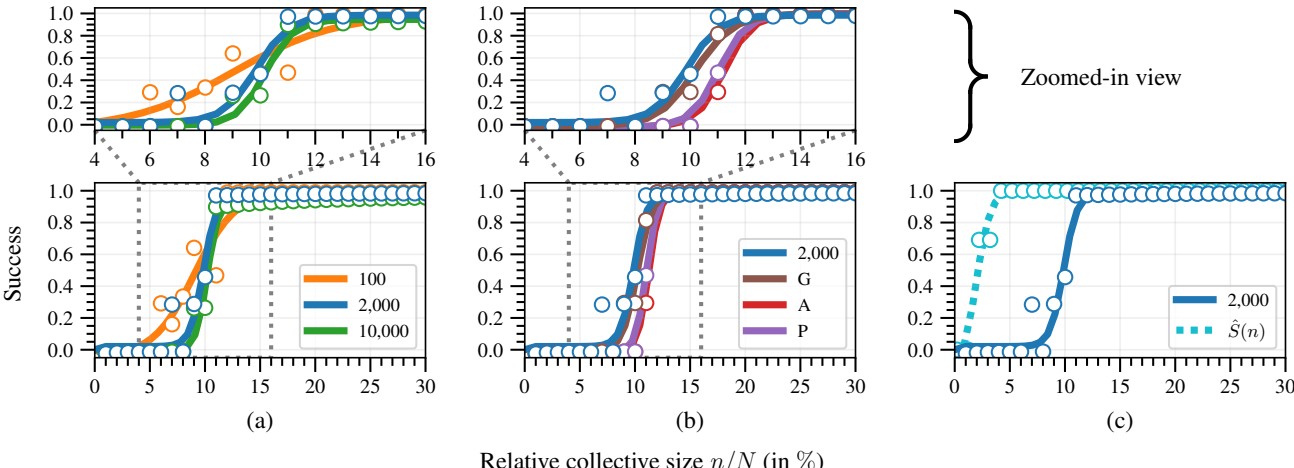

*Figure 3.* **Signal unplanting.** (a) Comparison of lower bounds from Theorem 3.7 for different values of $n_e$. (b) Comparison between the adaptive strategy with $n_e = 2,000$ and naive planting strategies targeting labels $y^* \in \{$Good (G), Average (A), Poor (P)$\}$. (c) Comparison of the lower bound achieved by the adaptive strategy with $n_e = 2,000$ and the actual success $\hat{S}(n)$ observed at test time.

The lower bound suggests that the collective would need to represent around $10\%$ of the agents interacting with the platform to have a significant impact, while in practice, only about $3\%$ was sufficient.

### 4.3. Comparative analysis

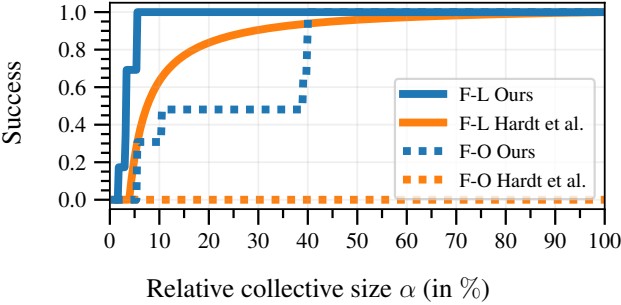

*Figure 4.* **Signal planting.** Comparison of signal planting lower bounds with target $y^* =$ Poor using feature-label (F-L) and feature-only (F-O) strategies. Our bounds in the infinite data regime are compared to bounds from Hardt et al. (2023), when $\varepsilon = 0$.

Our finite-sample framework differs from that presented by Hardt et al. (2023), which focuses on a population-level analysis. The connection is that the limit of our framework in the infinite data regime boils down to that of Hardt et al. (2023). Specifically, for $\alpha \in (0, 1)$, let $n, N, N_{\text{test}} \to \infty$ such that $n/N \to \alpha$. In this case, our statistical algorithmic collective action framework simplifies to that of algorithmic collective action. In Figure 4, we compare the bounds we obtain under the infinite data regime with those of Hardt et al. (2023). The bounds presented in our work are tighter than those obtained by Hardt et al. (2023). We provide a proof and additional remarks in Appendix F.

Interestingly, in the case of a discrete signal set, as used in our experiments, our lower bounds take on a *staircase* shape rather than a smooth sigmoid. This mirrors the shape of the curves observed in signal unplanting in Figure 3, which would also appear in Figure 1 and Figure 2 if we used finer increments of $n$. In contrast, the bounds from Hardt et al. (2023) do not capture this phenomenon.

## 5. Discussion

In this work, we introduced a framework where collectives aiming to influence a platform can leverage their local information by pooling their data. Our approach captures the key desideratum that collectives may not only want to observe the outcomes of their actions at test time but also anticipate lower bounds on their success. Moreover, it allows the collective to implement practical strategies based on parameters they do not directly observe, by using statistical estimation. The ability to anticipate outcomes and make informed decisions based on pooled data represents an advancement in understanding how collectives can interact with and influence platforms. However, there is room for further exploration.

We used Hoeffding's inequality as a proof of concept, but it is worth studying improved concentration inequalities for obtaining estimates of success for collectives. It is also possible to reverse the concentration inequalities to derive upper bounds on success, but the challenge lies in the fact that these upper bounds are often trivial. In ongoing work, we are exploiting recent developments in concentration inequalities due to Howard et al. (2020).

We focused on classification, but extensions to other objectives, such as regression, would be useful. We also assumed

that the feature space $\mathcal{X}$ is finite to use a limited number of events in the union bounds. It would be natural to remove this assumption by using a covering of $\tilde{\mathcal{X}}$, provided it is compact, to derive more general bounds for signal planting and unplanting.

Finally, a key assumption in collective action is that individuals are identically distributed. However, those who join a collective are often distributed differently from the general population—e.g., a collective against SUVs is likely to have fewer SUV-related data points and more data on smaller vehicles. Extending the framework to account for a heterogeneous population is an important direction for future work.

## Acknowledgements

The authors would like to thank Eugène Berta, Sacha Braun, Yurong Chen, and David Holzmüller for their helpful comments and suggestions.

Funded by the European Union (ERC-2022-SYG-OCEAN-101071601). Views and opinions expressed are however those of the author(s) only and do not necessarily reflect those of the European Union or the European Research Council Executive Agency. Neither the European Union nor the granting authority can be held responsible for them.

This publication is part of the Chair "Markets and Learning", supported by Air Liquide, BNP PARIBAS ASSET MANAGEMENT Europe, EDF, Orange and SNCF, sponsors of the Inria Foundation.

This work has also received support from the French government, managed by the National Research Agency, under the France 2030 program with the reference "PR[AI]RIE-PSAI" (ANR-23-IACL-0008).

## Impact Statement

This paper provides a theoretical and algorithmic foundation for studying collective influence, shedding light on the computations and coordination strategies necessary for such actions. By formalizing these mechanisms, we contribute to a deeper understanding of the risks and opportunities associated with collective interventions. This knowledge is essential for designing more robust, transparent, and socially aligned learning systems that account for strategic behavior and ensure fairer, more predictable outcomes in algorithmic decision-making.

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

# A. Additional Details on Experiments

We generate a dataset of 3,000,000 instances. Each instance represents a car, characterized by multiple categorical features that capture key aspects of vehicle design, performance, and manufacturing.

The features included in this synthetic dataset are as follows:

- *Model Type*: the type of vehicle, categorized as Sedan, SUV, Coupe, Hatchback, Convertible, Wagon, Minivan, or Truck.

- *Fuel Type*: the fuel used by the vehicle, which can be Gasoline, Diesel, Electric, or Hybrid.

- *Transmission Type*: the type of transmission, classified as Manual, Automatic, or CVT.

- *Drive Type*: the drive configuration of the vehicle, identified as FWD (Front-Wheel Drive), RWD (Rear-Wheel Drive), or AWD (All-Wheel Drive).

- *Safety Rating*: the safety rating of the vehicle, rated from 1 star to 5 stars.

- *Interior Material*: the material used for the vehicle's interior, which can be Cloth, Leather, or Synthetic.

- *Infotainment System*: the level of the infotainment system, ranging from Basic, Advanced, Premium, or None.

- *Country of Manufacture*: the country where the vehicle was manufactured, that we will denote by C1, C2, C3, C4, and C5.

- *Warranty Length*: the length of the vehicle's warranty, available in options of 3 years, 5 years, 7 years, or 10 years.

- *Number of Doors*: the number of doors on the vehicle, which can be 2, 4, or 5.

- *Number of Seats*: The seating capacity of the vehicle, with options for 2, 4, 5, or 7 seats.

- *Air Conditioning*: indicates whether the vehicle is equipped with air conditioning (Yes or No).

- *Navigation System*: the level of the navigation system, which can be None, Basic, or Advanced.

- *Tire Type*: the type of tires used, categorized as All-Season, Summer, or Winter.

- *Sunroof*: indicates whether the vehicle has a sunroof (Yes or No).

- *Sound System*: the quality of the sound system, which can be Standard, Premium, High-end, or None.

- *Cruise Control*: indicates whether the vehicle is equipped with cruise control (Yes or No).

- *Bluetooth Connectivity*: indicates whether the vehicle has Bluetooth connectivity (Yes or No).

Additionally, each car is assigned a *Car Evaluation* label based on a scoring system that considers various factors such as safety rating, fuel type, warranty length, and others. The possible evaluation outcomes are classified into four categories: Excellent, Good, Average, and Poor.

We generate separate consumer datasets, sampled without replacement from this base dataset. Unless otherwise specified, we choose $N = 1,000,000$ for the training set and $N_{\text{test}} = 100,000$ for the test set. In all the experiments, we set $\delta = 0.05$ and $\varepsilon = 0$.

In our experiments, the collective attempts to influence the platform by targeting features with specific characteristics defined through the transformation $g$: *Model Type* = SUV, *Fuel Type* = Diesel, *Transmission Type* = Manual, *Drive Type* = RWD, *Safety Rating* = 4 stars, *Interior Material* = Synthetic, *Infotainment System* = Premium, *Warranty Length* = 10 years, *Number of Doors* = 5, *Number of Seats* = 5, *Air Conditioning* = Yes, *Navigation System* = Advanced, *Tire Type* = All-Season, *Sunroof* = Yes, *Sound System* = Premium, *Cruise Control* = Yes, and *Bluetooth Connectivity* = Yes. In the dataset containing 3 million samples, the signal set $\tilde{\mathcal{X}}$ induced by $g$ comprises exactly 208,871 samples, which represents just under 7% of the entire training set. Table 1 provides details on the label frequencies within the signal set.

For further details on the dataset composition, we refer to the code available at: https://github.com/GauthierE/statistical-collusion.

*Table 1.* Distribution of labels for elements in $\tilde{\mathcal{X}}$, characterized by their *Country of Manufacture.*

| Country of Manufacture | Label | Sample Count |
|---|---|---|
| C1 | Excellent | 18410 |
| | Good | 9228 |
| | Average | 0 |
| | Poor | 1471 |
| C2 | Excellent | 18504 |
| | Good | 9214 |
| | Average | 0 |
| | Poor | 1461 |
| C3 | Excellent | 58083 |
| | Good | 29170 |
| | Average | 0 |
| | Poor | 4619 |
| C4 | Excellent | 17491 |
| | Good | 0 |
| | Average | 2946 |
| | Poor | 8911 |
| C5 | Excellent | 18589 |
| | Good | 9290 |
| | Average | 0 |
| | Poor | 1484 |
| Total | Excellent | 131077 |
| | Good | 56902 |
| | Average | 2946 |
| | Poor | 17946 |

# B. Algorithms

---

**Algorithm 1** Signal planting lower bound – feature-label strategy

---

1: Input: $\mathcal{X}, \mathcal{Y}, N, N_{\text{test}}, n < N, D^{(n)}, g, y^*, \delta > 0, \varepsilon > 0$
2: Define $\tilde{\mathcal{X}} := \{g(x) \mid x \in \mathcal{X}\}$
3: Observe $D^{(n)}$ and compute $\Delta_{\tilde{x}}^{(n)}$ for every $\tilde{x} \in \tilde{\mathcal{X}}$
4: Define $h : (x, y) \mapsto (g(x), y^*)$
5: Compute $\tilde{D}^{(n)}$ by applying $h$ to all samples in $D^{(n)}$
6: Compute $\hat{\mathbb{P}}_{\tilde{D}^{(n)}}(\tilde{x})$ for every $\tilde{x} \in \tilde{\mathcal{X}}$
7: Define $\tilde{\delta} := \delta/(2 + 2\#\tilde{\mathcal{X}} + 2\#\tilde{\mathcal{X}}\#\mathcal{Y})$
8: Compute $R_{\tilde{\delta}}(n)$, $R_{\tilde{\delta}}(N - n)$, and $R_{\tilde{\delta}}(N_{\text{test}})$
9: Compute and return:

$$\hat{\mathbb{P}}_{\tilde{x} \sim \tilde{D}^{(n)}} \left[ \frac{n}{N} \left( \hat{\mathbb{P}}_{\tilde{D}^{(n)}}(\tilde{x}) - 2R_{\tilde{\delta}}(n) \right) - \frac{N-n}{N} \left( \Delta_{\tilde{x}}^{(n)} + 2R_{\tilde{\delta}}(n) + 2R_{\tilde{\delta}}(N - n) \right) - \frac{\varepsilon}{1-\varepsilon} > 0 \right] - R_{\tilde{\delta}}(n) - R_{\tilde{\delta}}(N_{\text{test}})$$

---

---

**Algorithm 2** Signal planting lower bound – feature-only strategy

---

1: Input: $\mathcal{X}, \mathcal{Y}, N, N_{\text{test}}, n < N, D^{(n)}, g, y^*, \delta > 0, \varepsilon > 0$
2: Define $\tilde{\mathcal{X}} := \{g(x) \mid x \in \mathcal{X}\}$ and select some $x_0 \notin \tilde{\mathcal{X}}$
3: Observe $D^{(n)}$ and compute $\Delta_{\tilde{x}}^{(n)}$ for every $\tilde{x} \in \tilde{\mathcal{X}}$
4: Define $h : (x, y) \mapsto \begin{cases} (g(x), y^*) & \text{if } y = y^*, \\ (x_0, y) & \text{otherwise} \end{cases}$
5: Compute $\tilde{D}^{(n)}$ by applying $h$ to all samples in $D^{(n)}$
6: Compute $\hat{\mathbb{P}}_{\tilde{D}^{(n)}}(\tilde{x}, y^*)$ for every $\tilde{x} \in \tilde{\mathcal{X}}$
7: Define $\tilde{\delta} := \delta / (2 + 2\#\tilde{\mathcal{X}} + 2\#\tilde{\mathcal{X}}\#\mathcal{Y})$
8: Compute $R_{\tilde{\delta}}(n), R_{\tilde{\delta}}(N - n)$, and $R_{\tilde{\delta}}(N_{\text{test}})$
9: Compute and return:

$$\hat{\mathbb{P}}_{x' \sim D^{(n)}}\left[ \frac{n}{N}\left( \hat{\mathbb{P}}_{\tilde{D}^{(n)}}(g(x'), y^*) - 2R_{\tilde{\delta}}(n) \right) - \frac{N-n}{N}\left( \Delta_{g(x')}^{(n)} + 2R_{\tilde{\delta}}(n) + 2R_{\tilde{\delta}}(N-n) \right) - \frac{\varepsilon}{1-\varepsilon} > 0 \right] \quad - \quad R_{\tilde{\delta}}(n) \quad -$$
$$R_{\tilde{\delta}}(N_{\text{test}})$$

---

**Algorithm 3** Signal unplanting lower bound – adaptive strategy

---

1: Input: $\mathcal{X}, \mathcal{Y}, N, N_{\text{test}}, n < N, n_{\text{e}} < n, D^{(n_{\text{e}})}, D^{(n-n_{\text{e}})}, g, y^*, \delta > 0, \varepsilon > 0$
2: Define $\tilde{\mathcal{X}} := \{g(x) \mid x \in \mathcal{X}\}$
3: Observe $D^{(n_{\text{e}})}$ and compute $\hat{y}_{\tilde{x}} := \underset{y' \in \mathcal{Y} \setminus \{y^*\}}{\operatorname{argmax}} \hat{\mathbb{P}}_{D^{(n_{\text{e}})}}(\tilde{x}, y')$ for every $\tilde{x} \in \tilde{\mathcal{X}}$
4: Observe $D^{(n-n_{\text{e}})}$ and compute $\Delta_{\tilde{x}}^{(n-n_{\text{e}})}$ for every $\tilde{x} \in \tilde{\mathcal{X}}$
5: Define $h : (x, y) \mapsto (g(x), \hat{y}_{g(x)})$
6: Compute $\tilde{D}^{(n)}$ by applying $h$ to all samples in $D^{(n)}$ defined as the concatenation of $D^{(n_{\text{e}})}$ and $D^{(n-n_{\text{e}})}$
7: Compute $\hat{\mathbb{P}}_{\tilde{D}^{(n)}}(\tilde{x})$ for every $\tilde{x} \in \tilde{\mathcal{X}}$
8: Define $\tilde{\delta} := \delta / (2 + 6\#\tilde{\mathcal{X}})$
9: Compute $R_{\tilde{\delta}}(n), R_{\tilde{\delta}}(n - n_{\text{e}}), R_{\tilde{\delta}}(N - n)$, and $R_{\tilde{\delta}}(N_{\text{test}})$
10: Compute and return:

$$\hat{\mathbb{P}}_{\tilde{x} \sim \tilde{D}^{(n)}}\left[ \frac{n}{N}\left( \hat{\mathbb{P}}_{\tilde{D}^{(n)}}(\tilde{x}) - 2R_{\tilde{\delta}}(n) \right) - \frac{N-n}{N}\left( \Delta_{\tilde{x}}^{(n-n_{\text{e}})} + 2R_{\tilde{\delta}}(n - n_{\text{e}}) + 2R_{\tilde{\delta}}(N-n) \right) - \frac{\varepsilon}{1-\varepsilon} > 0 \right] \quad - \quad R_{\tilde{\delta}}(n) \quad -$$
$$R_{\tilde{\delta}}(N_{\text{test}})$$

---

**Algorithm 4** Signal erasing lower bound – under Assumption 1 and Assumption 2

---

1: Input: $\mathcal{X}, \mathcal{Y}, N, N_{\text{test}}, n < N, D^{(n)}, g, \delta > 0, \varepsilon > 0$
2: Define $\tilde{\mathcal{X}} := \{g(x) \mid x \in \mathcal{X}\}$
3: Observe $D^{(n)}$ and compute $y_{\tilde{x}}^* = \underset{y' \in \mathcal{Y}}{\operatorname{argmax}} \hat{\mathbb{P}}_{D^{(n)}}(\tilde{x}, y')$ for every $\tilde{x} \in \tilde{\mathcal{X}}$ and $\Delta_{x'}^{(n)}$ for every $x' \in \mathcal{X}$
4: Compute $\hat{\mathbb{P}}_{D^{(n)}}(x')$ for every $x' \in \mathcal{X}$
5: Define $\tilde{\delta} := \delta / (2 + \#\tilde{\mathcal{X}}\#\mathcal{Y} + 2\#\mathcal{X} + 2\#\mathcal{X}\#\mathcal{Y})$
6: Compute $R_{\tilde{\delta}}(n), R_{\tilde{\delta}}(N - n)$, and $R_{\tilde{\delta}}(N_{\text{test}})$
7: Compute and return:

$$\hat{\mathbb{P}}_{x' \sim D^{(n)}}\left[ \frac{n}{N}\left( \hat{\mathbb{P}}_{D^{(n)}}(x') - 2R_{\tilde{\delta}}(n) \right) - \frac{N-n}{N}\left( \Delta_{x'}^{(n)} + 2R_{\tilde{\delta}}(n) + 2R_{\tilde{\delta}}(N-n) \right) - \frac{\varepsilon}{1-\varepsilon} > 0 \right] - R_{\tilde{\delta}}(n) - R_{\tilde{\delta}}(N_{\text{test}})$$

---

Note that Algorithm 4 yields a valid lower bound if $\frac{2\log(1/\tilde{\delta})}{\eta^2} \leq n \leq N - \frac{2\log(1/\tilde{\delta})}{\eta^2}$ where $\eta$ is given in Assumption 1.

## C. Additional Experiments

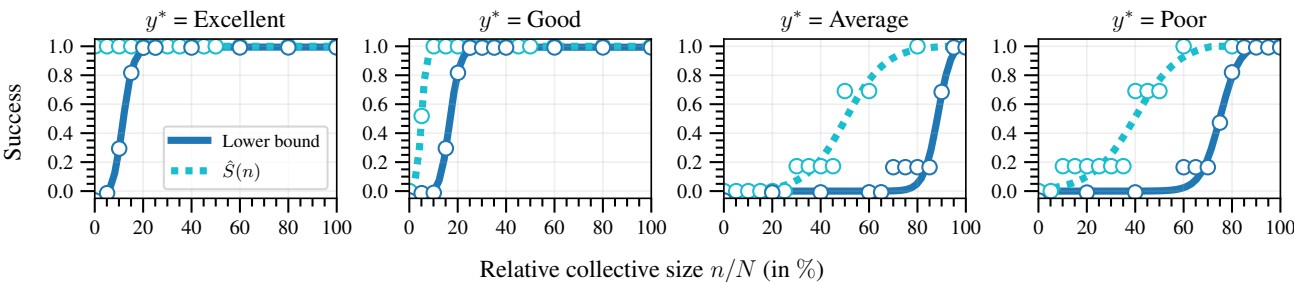

*Figure 5.* **Signal planting with feature-only strategy.** Comparison of the theoretical lower bound from Theorem 3.5 and the true success $\hat{S}(n)$ observed at test time for different values of $n$ and a fixed value of $N = 1,000,000$.

For the feature-only signal planting strategy, we selected $x_0$ constant equal to: *Model Type* = Sedan, *Fuel Type* = Diesel, *Transmission Type* = Automatic, *Drive Type* = RWD, *Safety Rating* = 1 star, *Interior Material* = Synthetic, *Infotainment System* = Premium, *Warranty Length* = 7 years, *Number of Doors* = 5, *Number of Seats* = 5, *Air Conditioning* = Yes, *Navigation System* = Advanced, *Tire Type* = All-Season, *Sunroof* = No, *Sound System* = Premium, *Cruise Control* = No, and *Bluetooth Connectivity* = No.

In Figure 5, we can see that both the lower bounds and the actual success rates at test time are lower compared to the feature-label strategy observed in Figure 1. This is expected since the collective has less leverage with the feature-only strategy, as it cannot modify its labels. Specifically, for planting $y^* = $ Average and $y^* = $ Poor, it appears that only collectives representing the majority of agents interacting with the learning platform are likely to have a significant impact, while the lower bounds remain uninformative unless the collective represents nearly all agents.

## D. Lemmas

**Lemma D.1** (Hoeffding's Inequality). *Let $X_1, X_2, \ldots, X_n$ be independent random variables such that $a_i \leq X_i \leq b_i$ for all $i$. Let $\overline{X} = \frac{1}{n} \sum_{i=1}^{n} X_i$. Then for any $t > 0$,*

$$\mathbb{P}_{X_1,\ldots,X_n} \left( \overline{X} - \mathbb{E}[\overline{X}] \geq t \right) \leq \exp\left( -\frac{2n^2 t^2}{\sum_{i=1}^{n} (b_i - a_i)^2} \right).$$

Note that by applying Lemma D.1 to the random variables $-X_i$ we obtain that:

$$\mathbb{P}_{X_1,\ldots,X_n} \left( \overline{X} - \mathbb{E}[\overline{X}] \leq -t \right) \leq \exp\left( -\frac{2n^2 t^2}{\sum_{i=1}^{n} (b_i - a_i)^2} \right).$$

**Lemma D.2** ((Hardt et al., 2023), Lemma B.1). *Suppose that $\mathcal{P}$ and $\mathcal{P}'$ are two distributions such that $TV(\mathcal{P}, \mathcal{P}') \leq \varepsilon$. Take any two events $E_1$ and $E_2$ measurable under $\mathcal{P}$ and $\mathcal{P}'$. If $\mathcal{P}(E_1) > \mathcal{P}(E_2) + \frac{\varepsilon}{1-\varepsilon}$, then $\mathcal{P}'(E_1) > \mathcal{P}'(E_2)$.*

## E. Proofs and additional remarks

### E.1. Proof of Theorem 3.3

*Proof.* The proof relies on applying union bound to several concentration inequalities. Here, we will use Hoeffding's inequality (Lemma D.1).

First, assume that $\varepsilon = 0$ for simplicity. Let $\tilde{x} \in \tilde{\mathcal{X}}$. By applying Hoeffding's inequality to the $n$ independent random variables $\mathbb{1}_{\{\tilde{x}'_i = \tilde{x}\}} \in [0,1]$ for $\tilde{x}'_i \in \tilde{D}^{(n)}$, we have that

$$\hat{\mathbb{P}}_{\tilde{x}' \sim \tilde{D}^{(n)}} (\tilde{x}' = \tilde{x}) \leq \mathbb{P}_{\tilde{x}' \sim \tilde{\mathcal{D}}} (\tilde{x}' = \tilde{x}) + R_{\tilde{\delta}}(n) \qquad (6)$$

with probability at least $1 - \tilde{\delta}$ over the draw of $D^{(n)}$. Similarly,

$$\mathbb{P}_{\tilde{x}' \sim \tilde{\mathcal{D}}} (\tilde{x}' = \tilde{x}) \leq \hat{\mathbb{P}}_{\tilde{x}' \sim \tilde{D}^{(n)}} (\tilde{x}' = \tilde{x}) + R_{\tilde{\delta}}(n) \qquad (7)$$

holds with probability at least $1 - \tilde{\delta}$ over the draw of $D^{(n)}$. For any fixed $\tilde{x} \in \tilde{\mathcal{X}}$ and $y' \in \mathcal{Y} \backslash \{y^*\}$, the following inequalities also hold individually with probability at least $1 - \tilde{\delta}$ over the draw of consumers:

$$\hat{\mathbb{P}}_{(x,y) \sim D^{(N-n)}} (x = \tilde{x}, y = y') \leq \mathbb{P}_{(x,y) \sim \mathcal{D}} (x = \tilde{x}, y = y') + R_{\tilde{\delta}}(N - n) \tag{8}$$

$$\mathbb{P}_{(x,y) \sim \mathcal{D}} (x = \tilde{x}, y = y^*) \leq \hat{\mathbb{P}}_{(x,y) \sim D^{(N-n)}} (x = \tilde{x}, y = y^*) + R_{\tilde{\delta}}(N - n) \tag{9}$$

Similarly:

$$\hat{\mathbb{P}}_{(x,y) \sim D^{(n)}} (x = \tilde{x}, y = y^*) \leq \mathbb{P}_{(x,y) \sim \mathcal{D}} (x = \tilde{x}, y = y^*) + R_{\tilde{\delta}}(n) \tag{10}$$

$$\mathbb{P}_{(x,y) \sim \mathcal{D}} (x = \tilde{x}, y = y') \leq \hat{\mathbb{P}}_{(x,y) \sim D^{(n)}} (x = \tilde{x}, y = y') + R_{\tilde{\delta}}(n) \tag{11}$$

We also have that

$$\hat{\mathbb{P}}_{x \sim D_{\text{test}}} (\hat{f}(g(x)) = y^*) \geq \mathbb{P}_{x \sim \mathcal{D}} (\hat{f}(g(x)) = y^*) - R_{\tilde{\delta}}(N_{\text{test}}) \tag{12}$$

with probability at least $1 - \tilde{\delta}$, where the probability is taken over $D_{\text{test}}$ conditionally on $D^{(n)}$ and $D^{(N-n)}$, so in particular it holds marginally over all consumers. Lastly, we have with probability at least $1 - \tilde{\delta}$ over the draw of $D^{(n)}$:

$$\hat{\mathbb{P}}_{\tilde{x} \sim \tilde{D}^{(n)}} \left( \frac{n}{N} \left( \mathbb{P}_{\tilde{x}' \sim \tilde{\mathcal{D}}} (\tilde{x}' = \tilde{x}) - R_{\tilde{\delta}}(n) \right) - \frac{N-n}{N} \left( \Delta_{\tilde{x}}^{(\mathcal{D})} + 2R_{\tilde{\delta}}(N - n) \right) > 0 \right) \leq$$
$$\mathbb{P}_{\tilde{x} \sim \tilde{\mathcal{D}}} \left( \frac{n}{N} \left( \mathbb{P}_{\tilde{x}' \sim \tilde{\mathcal{D}}} (\tilde{x}' = \tilde{x}) - R_{\tilde{\delta}}(n) \right) - \frac{N-n}{N} \left( \Delta_{\tilde{x}}^{(\mathcal{D})} + 2R_{\tilde{\delta}}(N - n) \right) > 0 \right) + R_{\tilde{\delta}}(n) \tag{13}$$

where $\Delta_{\tilde{x}}^{(\mathcal{D})} := \max_{y' \in \mathcal{Y} \backslash \{y^*\}} \left( \mathbb{P}_{(x,y) \sim \mathcal{D}} (x = \tilde{x}, y = y') - \mathbb{P}_{(x,y) \sim \mathcal{D}} (x = \tilde{x}, y = y^*) \right)$.

Now, we use union bound with these $2 + 2\#\tilde{\mathcal{X}} + 2\#\tilde{\mathcal{X}}\#\mathcal{Y}$ inequalities above, so that all the following calculations hold with probability at least $1 - \delta$ over the draw of consumers. First, Inequality (12) yields:

$$\hat{S}(n) \geq \mathbb{P}_{x \sim \mathcal{D}} (\hat{f}(g(x)) = y^*) - R_{\tilde{\delta}}(N_{\text{test}})$$
$$= \mathbb{P}_{\tilde{x} \sim \tilde{\mathcal{D}}} (\hat{f}(\tilde{x}) = y^*) - R_{\tilde{\delta}}(N_{\text{test}})$$

by definition of $h$. Now see that for any $\tilde{x} \in \tilde{\mathcal{X}}$ we have that

$$\hat{f}(\tilde{x}) = y^* \impliedby \forall y' \neq y^*, \hat{\mathcal{P}}(\tilde{x}, y^*) > \hat{\mathcal{P}}(\tilde{x}, y')$$
$$\iff \forall y' \neq y^*, \frac{n}{N} \hat{\mathbb{P}}_{(\tilde{x}', \tilde{y}') \sim \tilde{D}^{(n)}} (\tilde{x}' = \tilde{x}, \tilde{y}' = y^*) + \frac{N-n}{N} \hat{\mathbb{P}}_{(x,y) \sim D^{(N-n)}} (x = \tilde{x}, y = y^*)$$
$$> \frac{n}{N} \hat{\mathbb{P}}_{(\tilde{x}', \tilde{y}') \sim \tilde{D}^{(n)}} (\tilde{x}' = \tilde{x}, \tilde{y}' = y') + \frac{N-n}{N} \hat{\mathbb{P}}_{(x,y) \sim D^{(N-n)}} (x = \tilde{x}, y = y')$$
$$\iff \forall y' \neq y^*, \frac{n}{N} \hat{\mathbb{P}}_{\tilde{x}' \sim \tilde{D}^{(n)}} (\tilde{x}' = \tilde{x}) + \frac{N-n}{N} \hat{\mathbb{P}}_{(x,y) \sim D^{(N-n)}} (x = \tilde{x}, y = y^*)$$
$$> \frac{N-n}{N} \hat{\mathbb{P}}_{(x,y) \sim D^{(N-n)}} (x = \tilde{x}, y = y')$$
$$\iff \frac{n}{N} \hat{\mathbb{P}}_{\tilde{x}' \sim \tilde{D}^{(n)}} (\tilde{x}' = \tilde{x})$$
$$- \frac{N-n}{N} \max_{y' \in \mathcal{Y} \backslash \{y^*\}} \left( \hat{\mathbb{P}}_{(x,y) \sim D^{(N-n)}} (x = \tilde{x}, y = y') - \hat{\mathbb{P}}_{(x,y) \sim D^{(N-n)}} (x = \tilde{x}, y = y^*) \right) > 0$$
$$\impliedby \frac{n}{N} \left( \mathbb{P}_{\tilde{x}' \sim \tilde{\mathcal{D}}} (\tilde{x}' = \tilde{x}) - R_{\tilde{\delta}}(n) \right) - \frac{N-n}{N} \left( \Delta_{\tilde{x}}^{(\mathcal{D})} + 2R_{\tilde{\delta}}(N - n) \right) > 0$$

where the last implication comes from Inequality (7) and because

$$\max_{y'\in\mathcal{Y}\backslash\{y^*\}} \left( \hat{\mathbb{P}}_{(x,y)\sim D^{(N-n)}}(x=\tilde{x},y=y') - \hat{\mathbb{P}}_{(x,y)\sim D^{(N-n)}}(x=\tilde{x},y=y^*) \right)$$

$$\leq \max_{y'\in\mathcal{Y}\backslash\{y^*\}} \left( \mathbb{P}_{(x,y)\sim\mathcal{D}}(x=\tilde{x},y=y') - \mathbb{P}_{(x,y)\sim\mathcal{D}}(x=\tilde{x},y=y^*) \right) + 2R_{\tilde{\delta}}(N-n)$$

$$= \Delta_{\tilde{x}}^{(\mathcal{D})} + 2R_{\tilde{\delta}}(N-n)$$

by Inequality (8) on $\tilde{\mathcal{X}}\times\mathcal{Y}\backslash\{y^*\}$ and Inequality (9) on $\tilde{\mathcal{X}}\times\{y^*\}$. Therefore, we have that:

$$\hat{S}(n) \geq \mathbb{P}_{\tilde{x}\sim\tilde{\mathcal{D}}} \left( \frac{n}{N} \left( \mathbb{P}_{\tilde{x}'\sim\tilde{\mathcal{D}}}(\tilde{x}'=\tilde{x}) - R_{\tilde{\delta}}(n) \right) - \frac{N-n}{N} \left( \Delta_{\tilde{x}}^{(\mathcal{D})} + 2R_{\tilde{\delta}}(N-n) \right) > 0 \right)$$

$$- R_{\tilde{\delta}}(N_{\text{test}})$$

and so by applying Inequality (13) we deduce that:

$$\hat{S}(n) \geq \hat{\mathbb{P}}_{\tilde{x}\sim\tilde{D}^{(n)}} \left( \frac{n}{N} \left( \mathbb{P}_{\tilde{x}'\sim\tilde{\mathcal{D}}}(\tilde{x}'=\tilde{x}) - R_{\tilde{\delta}}(n) \right) - \frac{N-n}{N} \left( \Delta_{\tilde{x}}^{(\mathcal{D})} + 2R_{\tilde{\delta}}(N-n) \right) > 0 \right)$$

$$- R_{\tilde{\delta}}(n) - R_{\tilde{\delta}}(N_{\text{test}}).$$

Finally, note that:

$$\Delta_{\tilde{x}}^{(\mathcal{D})} = \max_{y'\in\mathcal{Y}\backslash\{y^*\}} \left( \mathbb{P}_{(x,y)\sim\mathcal{D}}(x=\tilde{x},y=y') - \mathbb{P}_{(x,y)\sim\mathcal{D}}(x=\tilde{x},y=y^*) \right)$$

$$\leq \max_{y'\in\mathcal{Y}\backslash\{y^*\}} \left( \hat{\mathbb{P}}_{(x,y)\sim D^{(n)}}(x=\tilde{x},y=y') - \hat{\mathbb{P}}_{(x,y)\sim D^{(n)}}(x=\tilde{x},y=y^*) \right) + 2R_{\tilde{\delta}}(n)$$

$$= \Delta_{\tilde{x}}^{(n)} + 2R_{\tilde{\delta}}(n)$$

by Inequality (10) on $\tilde{\mathcal{X}}\times\{y^*\}$ and Inequality (11) on $\tilde{\mathcal{X}}\times\mathcal{Y}\backslash\{y^*\}$. So together with Inequality (6), we conclude that:

$$\hat{S}(n) \geq \hat{\mathbb{P}}_{\tilde{x}\sim\tilde{D}^{(n)}} \left( \frac{n}{N} \left( \hat{\mathbb{P}}_{\tilde{x}'\sim\tilde{D}^{(n)}}(\tilde{x}'=\tilde{x}) - 2R_{\tilde{\delta}}(n) \right) - \frac{N-n}{N} \left( \Delta_{\tilde{x}}^{(n)} + 2R_{\tilde{\delta}}(n) + 2R_{\tilde{\delta}}(N-n) \right) > 0 \right)$$

$$- R_{\tilde{\delta}}(n) - R_{\tilde{\delta}}(N_{\text{test}}).$$

The case $\varepsilon > 0$ is similar, noting that for any $\tilde{x}\in\tilde{\mathcal{X}}$:

$$\hat{f}(\tilde{x}) = y^* \Longleftarrow \forall y' \neq y^*, \tilde{\mathcal{P}}(\tilde{x},y^*) > \tilde{\mathcal{P}}(\tilde{x},y')$$

$$\Longleftarrow \forall y' \neq y^*, \hat{\mathcal{P}}(\tilde{x},y^*) > \hat{\mathcal{P}}(\tilde{x},y') + \frac{\varepsilon}{1-\varepsilon} \quad \text{by Lemma D.2}$$

and with slight modifications to Inequality (13) to incorporate the term $\frac{\varepsilon}{1-\varepsilon}$. $\qquad\square$

If there exists a subset $\mathcal{X}_0 \subseteq \tilde{\mathcal{X}}$ such that $\#\mathcal{X}_0 < \#\tilde{\mathcal{X}}$ and $\mathbb{P}_{\tilde{x}\sim\tilde{\mathcal{D}}}(\tilde{x}\in\mathcal{X}_0)$ is significant, it may be preferable to condition the probability based on whether or not the feature belongs to $\mathcal{X}_0$, and to disregard the term conditioned on not being in $\mathcal{X}_0$. In this case, the union bound should be applied only to $\tilde{x}\in\mathcal{X}_0$.

The proof of Theorem 3.5 is essentially the same as the proof of Theorem 3.3. One difference is that we can not simplify $\mathbb{P}_{x\sim\mathcal{D}}(\hat{f}(g(x))=y^*) = \mathbb{P}_{\tilde{x}\sim\tilde{\mathcal{D}}}(\hat{f}(\tilde{x})=y^*)$ as we did in the proof of Theorem 3.3. Therefore, the calculations must be carried out directly using the term $g(x)$.

### E.2. Proof of Theorem 3.7

*Proof.* The proof closely follows that of Theorem 3.3, so we will only highlight the main differences. We focus on the case where $\varepsilon = 0$; the case where $\varepsilon > 0$ can be handled similarly to the proof of Theorem 3.3.

Let $\tilde{x} \in \tilde{\mathcal{X}}$. We have:

$$
\begin{aligned}
\hat{f}(\tilde{x}) \neq y^* &\Longleftarrow \exists y' \neq y^* \colon \hat{\mathcal{P}}(\tilde{x}, y^*) < \hat{\mathcal{P}}(\tilde{x}, y') \\
&\Longleftarrow \hat{\mathcal{P}}(\tilde{x}, y^*) < \hat{\mathcal{P}}(\tilde{x}, \hat{y}_{\tilde{x}}) \\
&\Longleftrightarrow \frac{n}{N} \underbrace{\hat{\mathbb{P}}_{(\tilde{x}', \tilde{y}') \sim \tilde{D}^{(n)}}(\tilde{x}' = \tilde{x}, \tilde{y}' = y^*)}_{=0} + \frac{N-n}{N} \hat{\mathbb{P}}_{(x,y) \sim D^{(N-n)}}(x = \tilde{x}, y = y^*) \\
&\qquad < \frac{n}{N} \underbrace{\hat{\mathbb{P}}_{(\tilde{x}', \tilde{y}') \sim \tilde{D}^{(n)}}(\tilde{x}' = \tilde{x}, \tilde{y}' = \hat{y}_{\tilde{x}})}_{= \hat{\mathbb{P}}_{\tilde{x}' \sim \tilde{D}^{(n)}}(\tilde{x}' = \tilde{x})} + \frac{N-n}{N} \hat{\mathbb{P}}_{(x,y) \sim D^{(N-n)}}(x = \tilde{x}, y = \hat{y}_{\tilde{x}}) \\
&\Longleftrightarrow \frac{n}{N} \hat{\mathbb{P}}_{\tilde{x}' \sim \tilde{D}^{(n)}}(\tilde{x}' = \tilde{x}) \\
&\qquad - \frac{N-n}{N} \underbrace{\left( \hat{\mathbb{P}}_{(x,y) \sim D^{(N-n)}}(x = \tilde{x}, y = y^*) - \hat{\mathbb{P}}_{(x,y) \sim D^{(N-n)}}(x = \tilde{x}, y = \hat{y}_{\tilde{x}}) \right)}_{=: \Delta_{\tilde{x}}^{(N-n)}} > 0
\end{aligned}
$$

and see that

$$
\Delta_{\tilde{x}}^{(N-n)} \leq \underbrace{\mathbb{P}_{(x,y) \sim \mathcal{D}}(x = \tilde{x}, y = y^*) - \mathbb{P}_{(x,y) \sim \mathcal{D}}(x = \tilde{x}, y = \hat{y}_{\tilde{x}})}_{=: \Delta_{\tilde{x}}^{(\mathcal{D})}} + 2R_{\tilde{\delta}}(N - n)
$$

by applying Hoeffding's inequality to both empirical probabilities. Note that Hoeffding's inequality for the term $\mathbb{P}_{(x,y) \sim \mathcal{D}}(x = \tilde{x}, y = \hat{y}_{\tilde{x}})$ holds in probability over $D^{(N-n)}$ conditionally on $D^{(n_e)}$. It should only be applied $\#\tilde{\mathcal{X}}$ times here contrary to $\#\tilde{\mathcal{X}}(\#\mathcal{Y} - 1)$ times in signal planting because $\hat{y}_{\tilde{x}}$ is fixed conditionally on $D^{(n_e)}$. Then, since the inequality holds in probability over $D^{(N-n)}$ conditionally on $D^{(n_e)}$, in particular it holds marginally over all consumers.

The conclusion is similar to that of Theorem 3.3. We just need to apply the same argument as above to use Hoeffding's inequality: $\mathbb{P}_{(x,y) \sim \mathcal{D}}(x = \tilde{x}, y = \hat{y}_{\tilde{x}}) \geq \hat{\mathbb{P}}_{(x,y) \sim D^{(n-n_e)}}(x = \tilde{x}, y = \hat{y}_{\tilde{x}}) - R_{\tilde{\delta}}(n - n_e)$. $\qquad \square$

In fact, it is possible to obtain a slightly better bound than (4). Indeed, to bound $\Delta_{\tilde{x}}^{(\mathcal{D})}$, one can use Hoeffding's inequality on $D^{(n)}$ instead of $D^{(n-n_e)}$ for the term $\mathbb{P}_{(x,y) \sim \mathcal{D}}(x = \tilde{x}, y = y^*)$. We kept the inequality (4) as it is to simplify the expression of the bound and make the interpretation of the different terms easier.

### E.3. Interpretation of the bound from Theorem 3.7

As the signal planting lower bound (1), the signal unplanting lower bound (4) also depends on three distinct terms:

- $\frac{n}{N} \left( \hat{\mathbb{P}}_{\tilde{x}' \sim \tilde{D}^{(n)}}(\tilde{x}' = \tilde{x}) - 2R_{\tilde{\delta}}(n) \right)$

- $-\frac{N-n}{N} \left( \Delta_{\tilde{x}}^{(n-n_e)} + 2R_{\tilde{\delta}}(n - n_e) + 2R_{\tilde{\delta}}(N - n) \right)$

- $-\frac{\varepsilon}{1-\varepsilon}$

The first and third terms play the same role as in signal planting. The main difference here is the second term $-\frac{N-n}{N}\left(\Delta_{\tilde{x}}^{(n-n_{\mathrm{e}})} + 2R_{\tilde{\delta}}(n - n_{\mathrm{e}}) + 2R_{\tilde{\delta}}(N - n)\right)$. The term $\Delta_{\tilde{x}}^{(n-n_{\mathrm{e}})}$ represents how much more probable the target label $y^*$ is compared to other labels given a feature belonging to $\tilde{\mathcal{X}}$. If $y^*$ is much more probable than the other labels, it will be difficult to unplant the signal. Conversely, if $y^*$ is not the most probable, it will be easier to unplant the signal. The collective's plan relies on the choice of $n_{\mathrm{e}} < n$. If $n_{\mathrm{e}}$ is large, it will be more likely for the collective to select the best possible label given a modified feature $\tilde{x}$ to maximize the impact of the strategy $h$. However, if $n_{\mathrm{e}}$ is too large compared to $n$, the estimation error $2R_{\tilde{\delta}}(n - n_{\mathrm{e}})$ may become too significant. Therefore, there is a trade-off for the optimal choice of the value $n_{\mathrm{e}}$. This trade-off depends on the specific dataset being considered.

### E.4. Proof of Theorem 3.9

*Proof.* Assume that $\varepsilon = 0$. The case of $\varepsilon > 0$ is handled in the same way as in signal planting and signal unplanting. By Hoeffding's inequality, we know that the following inequalities each hold with probability at least $1 - \tilde{\delta}$ (over the draw of consumers), for some fixed $\tilde{x} \in \tilde{\mathcal{X}}$ and $y' \in \mathcal{Y}\backslash\{y_{\tilde{x}}^*\}$:

$$\mathop{\mathbb{P}}_{(x,y)\sim\tilde{\mathcal{D}}}(x = \tilde{x}, y = y_{\tilde{x}}^*) \leq \mathop{\hat{\mathbb{P}}}_{(x,y)\sim\tilde{D}^{(n)}}(x = \tilde{x}, y = y_{\tilde{x}}^*) + R_{\tilde{\delta}}(n) \tag{14}$$

$$\mathop{\hat{\mathbb{P}}}_{(x,y)\sim\tilde{D}^{(n)}}(x = \tilde{x}, y = y') \leq \mathop{\mathbb{P}}_{(x,y)\sim\tilde{\mathcal{D}}}(x = \tilde{x}, y = y') + R_{\tilde{\delta}}(n) \tag{15}$$

Similarly, for some fixed $x' \in \mathcal{X}$ and $y' \in \mathcal{Y}\backslash\{y_{g(x')}^*\}$:

$$\mathop{\hat{\mathbb{P}}}_{x\sim D^{(n)}}(x = x') \leq \mathop{\mathbb{P}}_{x\sim\mathcal{D}}(x = x') + R_{\tilde{\delta}}(n) \tag{16}$$

$$\mathop{\mathbb{P}}_{x\sim\mathcal{D}}(x = x') \leq \mathop{\hat{\mathbb{P}}}_{x\sim D^{(n)}}(x = x') + R_{\tilde{\delta}}(n) \tag{17}$$

$$\mathop{\hat{\mathbb{P}}}_{(x,y)\sim D^{(N-n)}}(x = x', y = y') \leq \mathop{\mathbb{P}}_{(x,y)\sim\mathcal{D}}(x = x', y = y') + R_{\tilde{\delta}}(N - n) \tag{18}$$

$$\mathop{\mathbb{P}}_{(x,y)\sim\mathcal{D}}(x = x', y = y_{g(x')}^*) \leq \mathop{\hat{\mathbb{P}}}_{(x,y)\sim D^{(N-n)}}(x = x', y = y_{g(x')}^*) + R_{\tilde{\delta}}(N - n) \tag{19}$$

$$\mathop{\hat{\mathbb{P}}}_{(x,y)\sim D^{(n)}}(x = x', y = y_{g(x')}^*) \leq \mathop{\mathbb{P}}_{(x,y)\sim\mathcal{D}}(x = x', y = y_{g(x')}^*) + R_{\tilde{\delta}}(n) \tag{20}$$

$$\mathop{\mathbb{P}}_{(x,y)\sim\mathcal{D}}(x = x', y = y') \leq \mathop{\hat{\mathbb{P}}}_{(x,y)\sim D^{(n)}}(x = x', y = y') + R_{\tilde{\delta}}(n) \tag{21}$$

We also have that

$$\mathop{\hat{\mathbb{P}}}_{x'\sim D_{\text{test}}}(\hat{f}(g(x')) = \hat{f}(x')) \geq \mathop{\mathbb{P}}_{x'\sim\mathcal{D}}(\hat{f}(g(x')) = \hat{f}(x')) - R_{\tilde{\delta}}(N_{\text{test}}) \tag{22}$$

and

$$\begin{aligned}
&\mathop{\hat{\mathbb{P}}}_{x'\sim D^{(n)}}\left(\frac{n}{N}\left(\mathop{\mathbb{P}}_{x\sim\mathcal{D}}(x = x') - R_{\tilde{\delta}}(n)\right) - \frac{N-n}{N}\left(\Delta_{x'}^{(\mathcal{D})} + 2R_{\tilde{\delta}}(N - n)\right) > 0\right) \leq \\
&\mathop{\mathbb{P}}_{x'\sim\mathcal{D}}\left(\frac{n}{N}\left(\mathop{\mathbb{P}}_{x\sim\mathcal{D}}(x = x') - R_{\tilde{\delta}}(n)\right) - \frac{N-n}{N}\left(\Delta_{x'}^{(\mathcal{D})} + 2R_{\tilde{\delta}}(N - n)\right) > 0\right) + R_{\tilde{\delta}}(n)
\end{aligned} \tag{23}$$

where $\Delta_{x'}^{(\mathcal{D})} := \max_{y'\in\mathcal{Y}\backslash\{y_{g(x')}^*\}}\left(\mathop{\mathbb{P}}_{(x,y)\sim\mathcal{D}}(x = x', y = y') - \mathop{\mathbb{P}}_{(x,y)\sim\mathcal{D}}(x = x', y = y_{g(x')}^*)\right)$.

We apply union bound to the $2 + \#\tilde{\mathcal{X}}\#\mathcal{Y} + 2\#\mathcal{X} + 2\#\mathcal{X}\#\mathcal{Y}$ inequalities above, so that the calculations below hold with probability at least $1 - \delta$.

Let $\tilde{x} \in \tilde{\mathcal{X}}$. By Assumption 1 we know that:

$$\mathop{\mathbb{P}}_{(x,y)\sim\mathcal{D}}(x = \tilde{x}, y = y_{\tilde{x}}^*) > \mathop{\mathbb{P}}_{(x,y)\sim\mathcal{D}}(x = \tilde{x}, y = y') + \eta \tag{24}$$

for all $y' \neq y_{\tilde{x}}^*$. So by using Inequality (14) and Inequality (15):

$$\hat{\mathbb{P}}_{(x,y)\sim D^{(n)}}(x = \tilde{x}, y = y_{\tilde{x}}^*) + R_{\tilde{\delta}}(n) > \hat{\mathbb{P}}_{(x,y)\sim D^{(n)}}(x = \tilde{x}, y = y') - R_{\tilde{\delta}}(n) + \eta$$

for all $\forall y' \neq y_{\tilde{x}}^*$. Provided that $n \geq \frac{2\log(1/\tilde{\delta})}{\eta^2}$, we have that $\eta - 2R_{\delta}(n) > 0$ and so we deduce that

$$\hat{\mathbb{P}}_{(x,y)\sim D^{(n)}}(x = \tilde{x}, y = y_{\tilde{x}}^*) > \hat{\mathbb{P}}_{(x,y)\sim D^{(n)}}(x = \tilde{x}, y = y')$$

for all $\forall y' \neq y_{\tilde{x}}^*$.

Therefore, for each $\tilde{x} \in \tilde{\mathcal{X}}$, the collective can compute $y_{\tilde{x}}^*$ by using $D^{(n)}$ as follows:

$$y_{\tilde{x}}^* = \underset{y'\in\mathcal{Y}}{\operatorname{argmax}} \; \hat{\mathbb{P}}_{(x,y)\sim D^{(n)}}(x = \tilde{x}, y = y'). \tag{25}$$

Then, the collective plays the erasure stategy $h(x, y) = (x, y_{g(x)}^*)$.

Now see that using Inequality (18) and Inequality (19) together with Inequality (24) yields that

$$\hat{\mathbb{P}}_{(x,y)\sim D^{(N-n)}}(x = \tilde{x}, y = y_{\tilde{x}}^*) + R_{\tilde{\delta}}(N - n) > \hat{\mathbb{P}}_{(x,y)\sim D^{(N-n)}}(x = \tilde{x}, y = y') - R_{\tilde{\delta}}(N - n) + \eta$$

for all $y' \neq y_{\tilde{x}}^*$. Provided that $n \leq N - \frac{2\log(1/\tilde{\delta})}{\eta^2}$, we have that $\eta - 2R_{\tilde{\delta}}(N - n) > 0$ and so we deduce that

$$\hat{\mathbb{P}}_{(x,y)\sim D^{(N-n)}}(x = \tilde{x}, y = y_{\tilde{x}}^*) > \hat{\mathbb{P}}_{(x,y)\sim D^{(N-n)}}(x = \tilde{x}, y = y')$$

for all $\forall y' \neq y_{\tilde{x}}^*$. This implies that

$$\underset{y'\in\mathcal{Y}}{\operatorname{argmax}} \; \hat{\mathbb{P}}_{(x,y)\sim D^{(N-n)}}(x = \tilde{x}, y = y') = y_{\tilde{x}}^*. \tag{26}$$

So for all $x' \in \mathcal{X}$ we have:

$$y_{g(x')}^* = \underset{y'\in\mathcal{Y}}{\operatorname{argmax}} \left\{ \frac{n}{N} \underbrace{\hat{\mathbb{P}}_{(\tilde{x},\tilde{y})\sim\tilde{D}^{(n)}}(\tilde{x} = g(x'), \tilde{y} = y')}_{=0 \text{ if } y' \neq y_{g(x')}^* \text{ since } g(g(x')) = g(x')} + \frac{N-n}{N} \underbrace{\hat{\mathbb{P}}_{(x,y)\sim D^{(N-n)}}(x = g(x'), y = y')}_{\text{maximized in } y_{g(x')}^* \text{ by Equation (26)}} \right\}$$

$$= \underset{y'\in\mathcal{Y}}{\operatorname{argmax}} \; \hat{\mathcal{P}}(g(x'), y')$$

$$= \hat{f}(g(x')).$$

Therefore

$$\hat{S}(n) = \hat{\mathbb{P}}_{x'\sim D_{\text{test}}} \left( \hat{f}(x') = \hat{f}(g(x')) \right)$$

$$\geq \mathbb{P}_{x'\sim\mathcal{D}} \left( \hat{f}(x') = \hat{f}(g(x')) \right) - R_{\tilde{\delta}}(N_{\text{test}}) \quad \text{by Inequality (22)}$$

$$= \mathbb{P}_{x'\sim\mathcal{D}} \left( \hat{f}(x') = y_{g(x')}^* \right) - R_{\tilde{\delta}}(N_{\text{test}}).$$

Now let $x' \in \mathcal{X}$. A calculation similar to the one in the proof of Theorem 3.3 shows that

$$\hat{f}(x') = y_{g(x')}^* \Longleftarrow \frac{n}{N} \left( \mathbb{P}_{x\sim\mathcal{D}}(x = x') - R_{\tilde{\delta}}(n) \right) - \frac{N-n}{N} \left( \Delta_{x'}^{(\mathcal{D})} + 2R_{\tilde{\delta}}(N - n) \right) > 0.$$

Here we reused Inequality (18) and Inequality (19), together with Inequality (17) and the fact that $\tilde{\mathcal{D}}_{\mathcal{X}} = \mathcal{D}_{\mathcal{X}}$ by definition of $h$.

Therefore

$$\hat{S}(n) \geq \underset{x' \sim \mathcal{D}}{\mathbb{P}} \left( \frac{n}{N} \left( \underset{x \sim \mathcal{D}}{\mathbb{P}}(x = x') - R_{\tilde{\delta}}(n) \right) - \frac{N-n}{N} \left( \Delta_{x'}^{(\mathcal{D})} + 2R_{\tilde{\delta}}(N-n) \right) > 0 \right)$$
$$- R_{\tilde{\delta}}(N_{\text{test}})$$

and so by applying Inequality (23) we deduce that

$$\hat{S}(n) \geq \underset{x' \sim D^{(n)}}{\mathbb{P}} \left( \frac{n}{N} \left( \underset{x \sim \mathcal{D}}{\mathbb{P}}(x = x') - R_{\tilde{\delta}}(n) \right) - \frac{N-n}{N} \left( \Delta_{x'}^{(\mathcal{D})} + 2R_{\tilde{\delta}}(N-n) \right) > 0 \right)$$
$$- R_{\tilde{\delta}}(n) - R_{\tilde{\delta}}(N_{\text{test}}).$$

We conclude with Inequality (16), Inequality (20), and Inequality (21). $\qquad \square$

The result of Theorem 3.9 depends on whether the collective is large enough to compute the optimal label $y_{\tilde{x}}^*$ for each $\tilde{x} \in \tilde{\mathcal{X}}$. Specifically, $n$ must exceed $\frac{2\log(1/\tilde{\delta})}{\eta^2}$. In Figure 6, we plot $\frac{2\log(1/\tilde{\delta})}{\eta^2}$ as a function of $\eta$ with the parameters used in our experiment in Section 4 and detailed in Appendix A: $\#\mathcal{X} = 2,388,787,200$; $\#\tilde{\mathcal{X}} = 5$; and $\#\mathcal{Y} = 4$.

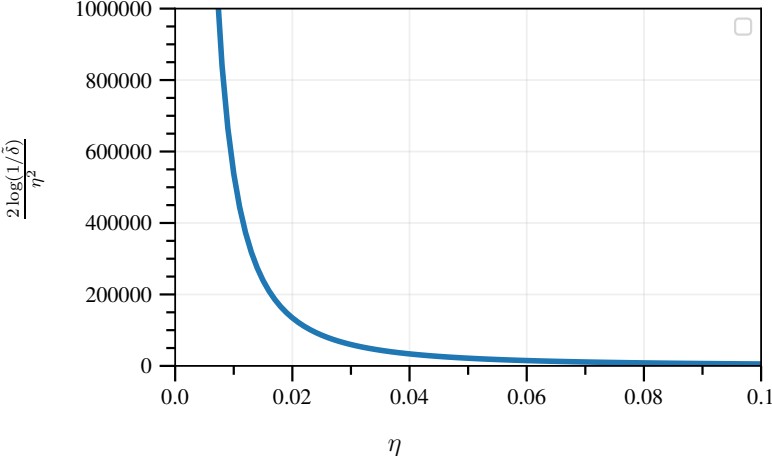

*Figure 6.* Plot of the minimum value of $n$ as a function of $\eta$ for computing the erasure strategy.

For example, suppose the collective consists of $n = 100,000$ members. Then, if $\eta$ exceeds approximately 0.03, the signal-erasing bound 5 from Theorem 3.9 holds.

### E.5. Interpretation of the bound from Theorem 3.9

The signal erasing lower bound 5 depends on three distinct terms:

- $\frac{n}{N} \left( \underset{x \sim D^{(n)}}{\hat{\mathbb{P}}}(x = x') - 2R_{\tilde{\delta}}(n) \right)$

- $-\frac{N-n}{N} \left( \Delta_{x'}^{(n)} + 2R_{\tilde{\delta}}(n) + 2R_{\tilde{\delta}}(N-n) \right)$

- $-\frac{\varepsilon}{1-\varepsilon}$

The second term captures how optimal the label for the feature $g(x)$ is for the feature $x$ as well, thus reflecting the initial sensitivity of the signal, which aligns with the intuition discussed by Hardt et al. (2023).

There are two major differences between signal erasing and signal planting or unplanting. First, the term $\frac{n}{N}\left(\hat{\underset{x \sim D^{(n)}}{\mathbb{P}}}(x = x') - 2R_{\tilde{\delta}}(n)\right)$ is significantly weaker in the case of signal erasing because it depends on $\mathcal{X}$ rather than $\tilde{\mathcal{X}}$, making the probability much lower. Second, the term $\tilde{\delta}$ is much smaller, making the estimation terms much more significant. In most cases, we can expect the obtained bound in Theorem 3.9 to be impractical because the estimation terms will be too large. The bound in Theorem 3.9 becomes useful only when we have a large number of data points relative to the size of the universe $\mathcal{X} \times \mathcal{Y}$.

We provide an analysis of the estimation terms to better understand their influence not only in signal erasing but also in signal planting and signal unplanting. We illustrate it with a basic example in Figure 7. We fix $\delta = 0.05$. We consider a binary classification problem: $\#\mathcal{Y} = 2$. In this case, the values of $\tilde{\delta}$ are as follows: $\delta/(2 + 6\#\tilde{\mathcal{X}})$ in signal planting and signal unplanting, and $\delta/(2 + 4\#\tilde{\mathcal{X}} + 6\#\mathcal{X}) \approx \delta/(2 + 6\#\mathcal{X})$ in signal erasing, assuming the reasonable hypothesis that $\#\tilde{\mathcal{X}} \ll \#\mathcal{X}$. Therefore, we focus on the estimation terms $R_{\tilde{\delta}}(n)$ with $\tilde{\delta}$ of the form $\delta/(2 + 6\gamma)$ where $\gamma$ varies. For simplicity, we assume that $\#\tilde{\mathcal{X}}$ and $\#\mathcal{X}$ are powers of two: $\gamma = 2^m$ for some $m \geq 1$.

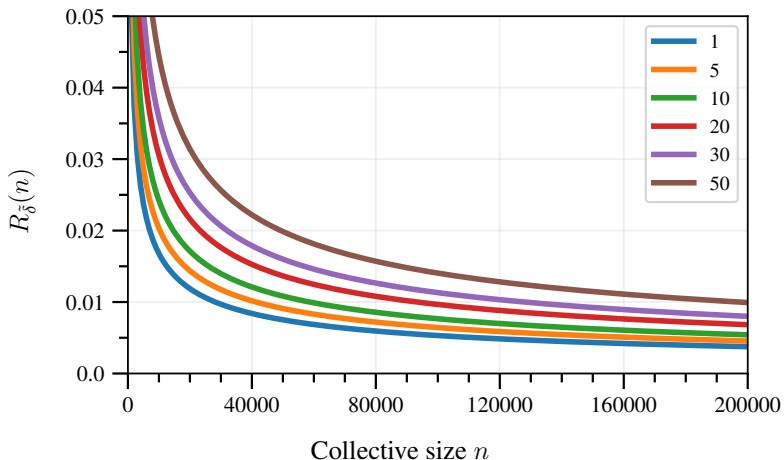

Figure 7. Plot of estimation terms $R_{\tilde{\delta}}(n)$ as a function of $n$ for various values of $m$, where $\tilde{\delta} = \delta/(2 + 6 \times 2^m)$.

Given a target error estimation $R_*$, the value $n$ at which $R_{\tilde{\delta}}(n) \leq R_*$ is approximately equal to $n \approx \frac{\log(2)}{2R_*^2}m + \frac{\log(6) + \log(1/\delta)}{2R_*^2}$. This value of $n$ scales linearly with $m$.

This simple analysis highlights an interesting phenomenon: the lower bounds obtained are influenced not only by the ratio $n/N$ (i.e., how large $n$ is relative to $N$) but also by the absolute size of $n$ in relation to the universe $\mathcal{X} \times \mathcal{Y}$. This is crucial for obtaining estimation terms $R_{\tilde{\delta}}(n)$ that are as small as possible. The intuition that $n$ needs to be sufficiently large relative to $N$ is already present in the work by Hardt et al. (2023). Our research sheds light on the additional requirement that $n$ should also be large in comparison to the size of the considered universe $\mathcal{X} \times \mathcal{Y}$.

## F. Bounds in the Infinite Data Regime

As discussed in Subsection 4.3, we can compare our results with those presented by Hardt et al. (2023) in the infinite data regime when $n, N, N_{\text{test}} \to \infty$ and $n/N \to \alpha$ for some $\alpha \in (0, 1)$. We provide further details on this comparison here.

In the infinite data regime, the framework is the same as in algorithmic collective action: the platform observes the distribution

$$\mathcal{P}(x_0, y_0) := \alpha \underset{\tilde{\mathcal{D}}}{\mathbb{P}}(x_0, y_0) + (1 - \alpha) \underset{\mathcal{D}}{\mathbb{P}}(x_0, y_0)$$

and selects a classifier $f$ based on the distribution $\mathcal{P}$:

**Definition F.1.** Let $\varepsilon > 0$. A classifier $f : \mathcal{X} \to \mathcal{Y}$ is $\varepsilon$-suboptimal on a set $\mathcal{X}' \subseteq \mathcal{X}$ under the distribution $\mathcal{P}$ if there exists a distribution $\tilde{\mathcal{P}}$ with $TV(\mathcal{P}, \tilde{\mathcal{P}}) \leq \varepsilon$ such that

$$f(x) \in \underset{y \in \mathcal{Y}}{\operatorname{argmax}} \tilde{\mathcal{P}}(x, y)$$

for all $x \in \mathcal{X}'$.

With these notations, we can redefine the successes for the various objectives of the collective and derive lower bounds on these successes. Note that to obtain the bounds below, it is no longer necessary to assume that $\mathcal{X}$ is finite, since this assumption is only used to apply the union bound a finite number of times when the collective performs statistical inference with its finite-sample dataset.

### F.1. Signal planting

The success is defined as

$$S(\alpha) := \mathop{\mathbb{P}}_{x \sim \mathcal{D}}(f(g(x)) = y^*).$$

The bound (1) from Theorem 3.3 under the feature-label strategy becomes:

$$S(\alpha) \geq \mathop{\mathbb{P}}_{\tilde{x} \sim \tilde{\mathcal{D}}} \left[ \alpha \mathop{\mathbb{P}}_{\tilde{\mathcal{D}}}(\tilde{x}) - (1 - \alpha)\Delta_{\tilde{x}}^{(\mathcal{D})} - \frac{\varepsilon}{1 - \varepsilon} > 0 \right] \tag{27}$$

where $\Delta_{\tilde{x}}^{(\mathcal{D})} = \max_{y' \in \mathcal{Y} \setminus \{y^*\}} \left( \mathop{\mathbb{P}}_{\mathcal{D}}(\tilde{x}, y') - \mathop{\mathbb{P}}_{\mathcal{D}}(\tilde{x}, y^*) \right)$.

Similarly, the bound (2) from Theorem 3.5 under the feature-only strategy becomes:

$$S(\alpha) \geq \mathop{\mathbb{P}}_{x \sim \mathcal{D}} \left[ \alpha \mathop{\mathbb{P}}_{\tilde{\mathcal{D}}}(g(x), y^*) - (1 - \alpha)\Delta_{g(x)}^{(\mathcal{D})} - \frac{\varepsilon}{1 - \varepsilon} > 0 \right]. \tag{28}$$

### F.2. Signal unplanting

The success is defined as

$$S(\alpha) := \mathop{\mathbb{P}}_{x \sim \mathcal{D}}(f(g(x)) \neq y^*).$$

The signal unplanting strategy relies on the choice of some parameter $n_e < n$ to estimate the most likely label $y_{g(x)} \neq y^*$ given some feature $g(x)$. In the infinite data regime, we assume that $n_e \to \infty$ and $n - n_e \to \infty$. The bound (4) from Theorem 3.7 becomes:

$$S(\alpha) \geq \mathop{\mathbb{P}}_{\tilde{x} \sim \tilde{\mathcal{D}}} \left[ \alpha \mathop{\mathbb{P}}_{\tilde{\mathcal{D}}}(\tilde{x}) - (1 - \alpha)\Delta_{\tilde{x}}^{(\mathcal{D})} - \frac{\varepsilon}{1 - \varepsilon} > 0 \right] \tag{29}$$

where $\Delta_{\tilde{x}}^{(\mathcal{D})} = \mathop{\mathbb{P}}_{\mathcal{D}}(\tilde{x}, y^*) - \mathop{\mathbb{P}}_{\mathcal{D}}(\tilde{x}, y_{\tilde{x}})$.

### F.3. Signal erasing

The success is defined as

$$S(\alpha) := \mathop{\mathbb{P}}_{x \sim \mathcal{D}}(f(g(x)) = f(x)).$$

The erasure strategy relies on the ability to compute the optimal label $y_{g(x)}^*$ for every feature $g(x)$. In the infinite data regime, we assume that the collective has access to $y_{g(x)}^*$. The bound (5) from Theorem 3.9 becomes:

$$S(\alpha) \geq \mathop{\mathbb{P}}_{x \sim \mathcal{D}} \left[ \alpha \mathop{\mathbb{P}}_{\mathcal{D}}(x) - (1 - \alpha)\Delta_x^{(\mathcal{D})} - \frac{\varepsilon}{1 - \varepsilon} > 0 \right] \tag{30}$$

where $\Delta_x^{(\mathcal{D})} := \max_{y' \in \mathcal{Y} \setminus \{y_{g(x)}^*\}} \mathop{\mathbb{P}}_{\mathcal{D}}(x, y') - \mathop{\mathbb{P}}_{\mathcal{D}}(x, y_{g(x)}^*)$.

### F.4. Comparison with Prior Lower Bounds

In the original version of their work, Hardt et al. (2023) adopt the same definition of $\varepsilon$-suboptimality as in our work. However, there was an issue with their proof in the case where $\varepsilon > 0$. For this reason, we restrict our comparison to the bounds in the case where $\varepsilon = 0$.

**Proposition F.2** (Bounds comparison in signal planting when $\varepsilon = 0$). *Assume that the collective plays the feature-label signal planting strategy against a classifier that is $\varepsilon$-suboptimal (in the sense of Definition F.1) on $\tilde{\mathcal{X}}$. Then the collective achieves:*

$$S(\alpha) \geq \mathop{\mathbb{P}}_{\tilde{x} \sim \tilde{\mathcal{D}}} \left[ \alpha \mathop{\mathbb{P}}_{\tilde{\mathcal{D}}}(\tilde{x}) - (1-\alpha) \Delta_{\tilde{x}}^{(\mathcal{D})} > 0 \right]$$

$$\geq 1 - \frac{1-\alpha}{\alpha} \mathop{\mathbb{P}}_{\mathcal{D}}\left(\tilde{\mathcal{X}}\right) \max_{\tilde{x} \in \tilde{\mathcal{X}}} \max_{y \in \mathcal{Y}} \left( \mathop{\mathbb{P}}_{\mathcal{D}}(y|\tilde{x}) - \mathop{\mathbb{P}}_{\mathcal{D}}(y^*|\tilde{x}) \right)$$

*where the first inequality is simply Inequality (27) and the last term is the lower bound originally provided by Hardt et al. (2023).*

*Proof.* For all $\tilde{x} \in \tilde{\mathcal{X}}$, we have:

$$\alpha \mathop{\mathbb{P}}_{\tilde{\mathcal{D}}}(\tilde{x}) - (1-\alpha) \Delta_{\tilde{x}}^{(\mathcal{D})} = \alpha \mathop{\mathbb{P}}_{\tilde{\mathcal{D}}}(\tilde{x}) - (1-\alpha) \max_{y' \in \mathcal{Y} \setminus \{y^*\}} \left( \mathop{\mathbb{P}}_{\mathcal{D}}(\tilde{x}, y') - \mathop{\mathbb{P}}_{\mathcal{D}}(\tilde{x}, y^*) \right)$$

$$= \alpha \mathop{\mathbb{P}}_{\tilde{\mathcal{D}}}(\tilde{x}) - (1-\alpha) \max_{y' \in \mathcal{Y} \setminus \{y^*\}} \left( \mathop{\mathbb{P}}_{\mathcal{D}}(y'|\tilde{x}) - \mathop{\mathbb{P}}_{\mathcal{D}}(y^*|\tilde{x}) \right) \mathop{\mathbb{P}}_{\mathcal{D}}(\tilde{x})$$

$$\geq \alpha \mathop{\mathbb{P}}_{\tilde{\mathcal{D}}}(\tilde{x}) - (1-\alpha) \max_{\tilde{x} \in \tilde{\mathcal{X}}} \max_{y' \in \mathcal{Y}} \left( \mathop{\mathbb{P}}_{\mathcal{D}}(y'|\tilde{x}) - \mathop{\mathbb{P}}_{\mathcal{D}}(y^*|\tilde{x}) \right) \mathop{\mathbb{P}}_{\mathcal{D}}(\tilde{x})$$

Therefore

$$\mathop{\mathbb{P}}_{\tilde{x} \sim \tilde{\mathcal{D}}} \left[ \alpha \mathop{\mathbb{P}}_{\tilde{\mathcal{D}}}(\tilde{x}) - (1-\alpha) \Delta_{\tilde{x}}^{(\mathcal{D})} > 0 \right] \geq \mathop{\mathbb{P}}_{\tilde{x} \sim \tilde{\mathcal{D}}} \left[ \alpha \mathop{\mathbb{P}}_{\tilde{\mathcal{D}}}(\tilde{x}) - (1-\alpha) \max_{\tilde{x} \in \tilde{\mathcal{X}}} \max_{y \in \mathcal{Y}} \left( \mathop{\mathbb{P}}_{\mathcal{D}}(y|\tilde{x}) - \mathop{\mathbb{P}}_{\mathcal{D}}(y^*|\tilde{x}) \right) \mathop{\mathbb{P}}_{\mathcal{D}}(\tilde{x}) > 0 \right]$$

$$= \mathop{\mathbb{P}}_{\tilde{x} \sim \tilde{\mathcal{D}}} \left[ 1 - \frac{1-\alpha}{\alpha} \max_{\tilde{x} \in \tilde{\mathcal{X}}} \max_{y \in \mathcal{Y}} \left( \mathop{\mathbb{P}}_{\mathcal{D}}(y|\tilde{x}) - \mathop{\mathbb{P}}_{\mathcal{D}}(y^*|\tilde{x}) \right) \frac{\mathop{\mathbb{P}}_{\mathcal{D}}(\tilde{x})}{\mathop{\mathbb{P}}_{\tilde{\mathcal{D}}}(\tilde{x})} > 0 \right]$$

$$= \mathop{\mathbb{E}}_{\tilde{x} \sim \tilde{\mathcal{D}}} \left[ \mathbb{1}_{\left\{ 1 - \frac{1-\alpha}{\alpha} \max_{\tilde{x} \in \tilde{\mathcal{X}}} \max_{y \in \mathcal{Y}} \left( \mathop{\mathbb{P}}_{\mathcal{D}}(y|\tilde{x}) - \mathop{\mathbb{P}}_{\mathcal{D}}(y^*|\tilde{x}) \right) \frac{\mathop{\mathbb{P}}_{\mathcal{D}}(\tilde{x})}{\mathop{\mathbb{P}}_{\tilde{\mathcal{D}}}(\tilde{x})} > 0 \right\}} \right]$$

$$\geq \mathop{\mathbb{E}}_{\tilde{x} \sim \tilde{\mathcal{D}}} \left[ 1 - \frac{1-\alpha}{\alpha} \max_{\tilde{x} \in \tilde{\mathcal{X}}} \max_{y \in \mathcal{Y}} \left( \mathop{\mathbb{P}}_{\mathcal{D}}(y|\tilde{x}) - \mathop{\mathbb{P}}_{\mathcal{D}}(y^*|\tilde{x}) \right) \frac{\mathop{\mathbb{P}}_{\mathcal{D}}(\tilde{x})}{\mathop{\mathbb{P}}_{\tilde{\mathcal{D}}}(\tilde{x})} \right]$$

$$= 1 - \frac{1-\alpha}{\alpha} \mathop{\mathbb{P}}_{\mathcal{D}}\left(\tilde{\mathcal{X}}\right) \max_{\tilde{x} \in \tilde{\mathcal{X}}} \max_{y \in \mathcal{Y}} \left( \mathop{\mathbb{P}}_{\mathcal{D}}(y|\tilde{x}) - \mathop{\mathbb{P}}_{\mathcal{D}}(y^*|\tilde{x}) \right)$$

$\square$

The authors subsequently revised their proof when $\varepsilon > 0$ by modifying the suboptimality definition, which we refer to as $\varepsilon$-conditional suboptimality to provide clarity and distinguish between the two definitions:

**Definition F.3.** Let $\varepsilon > 0$. A classifier $f : \mathcal{X} \to \mathcal{Y}$ is $\varepsilon$-conditionally suboptimal on a set $\mathcal{X}' \subseteq \mathcal{X}$ under the distribution $\mathcal{P}$ if there exists a distribution $\tilde{\mathcal{P}}$ with $TV(\mathcal{P}_{Y|X=x}, \tilde{\mathcal{P}}_{Y|X=x}) \leq \varepsilon$ such that

$$f(x) \in \operatorname*{argmax}_{y \in \mathcal{Y}} \tilde{\mathcal{P}}(x, y)$$

for all $x \in \mathcal{X}'$.

Using this definition of conditional suboptimality, we could derive bounds similar to those given in (1), (2), (4), (5), (27), (28), (29), and (30), and conduct the same kind of comparisons with the bounds from Hardt et al. (2023).

