# OpenReview forum: "Statistical Collusion by Collectives on Learning Platforms"
_ICML.cc/2025/Conference — ICML 2025 oral_

### Official Review · Reviewer_TeJJ · 2025-03-09

**Overall Recommendation:** 4

**Summary:**

This paper introduces a framework in which a group of users, or “collective,” can statistically coordinate modifications to their data to influence a platform’s learning algorithm. The authors define several types of objectives—“signal planting,” “signal unplanting,” and “signal erasing”—and provide theoretical guarantees on how effectively a collective can achieve these objectives. By modeling the platform as choosing an ε-suboptimal classifier, the work derives lower bounds on the collective’s success under finite-sample conditions. The paper proposes practical strategies for modifying data, shows how a collective can estimate crucial distributional parameters from its own pooled samples, and presents empirical results on a synthetic car-rating dataset. Overall, the paper contributes new algorithmic tools and tighter theoretical bounds that extend and refine earlier studies on collective action in machine learning.

**Claims And Evidence:**

Overall, the paper’s theoretical claims—namely that finite-sample data poisoning strategies can guarantee lower bounds for signal planting, unplanting, and erasing—are backed by proofs and validated with synthetic experiments. However, some broader practical claims are less firmly grounded:

Real-World Applicability: The paper relies on a synthetic car-rating dataset to illustrate the theory. While it demonstrates the feasibility of the strategies in a controlled setting, it does not provide equally detailed evidence that these methods scale or remain effective in complex, real-world data environments.

Platform Defenses: Although the paper discusses how a platform chooses an ε-suboptimal classifier based on the poisoned data, it does not deeply explore realistic detection or mitigation strategies. Consequently, any claim implying that these results readily extend to platforms with active defenses remains under-supported.

These limitations do not undermine the core theoretical findings, but they do mean that some broader statements regarding practical deployment would benefit from additional evidence or real-world trials to be wholly convincing.

**Essential References Not Discussed:**

NA

**Experimental Designs Or Analyses:**

I looked at how the synthetic car-rating dataset was generated and how success was measured for the different poisoning strategies (signal planting, unplanting, erasing). The experimental design aligns well with the theoretical framework—each experiment straightforwardly tests whether the derived lower bounds match observed outcomes on the synthetic data. While the dataset is simplified and may not fully capture real-world complexity, I did not see flaws in how the experiments were conducted or analyzed relative to the stated objectives.

**Methods And Evaluation Criteria:**

Yes. The paper’s methods—particularly the definitions of success in “signal planting,” “unplanting,” and “erasing”—align directly with the theoretical goals, and the synthetic car-rating dataset provides a controlled way to demonstrate how these objectives can be measured in practice. While real-world datasets might introduce more complexity, the chosen evaluation still makes sense as a proof-of-concept for the proposed finite-sample bounds and strategies.

**Other Comments Or Suggestions:**

Na

**Other Strengths And Weaknesses:**

Na

**Questions For Authors:**

Na

**Relation To Broader Scientific Literature:**

Compared to classic data poisoning, which often involves a single adversary, this work extends “collective action” ideas—originally from Hardt et al. (2023)—to finite-sample settings. It refines how groups of individuals coordinate data modifications (e.g., “signal planting,” “unplanting,” “erasing”) and integrates statistical estimation techniques so that the collective can learn optimal poisoning strategies from its own local data. This bridges backdoor and adversarial ML with fundamental estimation theory, yielding tighter and more broadly applicable bounds.

**Theoretical Claims:**

The arguments appear sound, and I did not spot errors in the manipulations or the final bounds. The stepwise presentation (outlining each concentration term, then taking a union bound) is internally consistent and aligns with standard proof techniques in learning theory.

---

> ### Author Rebuttal · Authors · 2025-03-28
>
> We thank the reviewer for their time and appreciate their positive feedback on our paper.

---

### Official Review · Reviewer_HD8o · 2025-03-10

**Overall Recommendation:** 4

**Summary:**

The paper proposes a framework to quantify the outcome of collective actions in machine learning, performed through the coordinated submission of altered data. In particular, the proposed framework allows the derivation of lower bounds on the success of the collective's strategy. The authors advocate for strategies that will be available in practice for a collective. They also advocate for the autonomy of the collective in estimating the success of a strategy based on information available to the collective. The framework focuses on three objectives, namely *signal planting*, *signal erasing*, and *signal unplanting* (newly introduced by the authors), and proposes efficient strategies to implement them, as well as lower bounds on their success.

For the experimental aspect of the work, the authors focused on *signal planting* and *signal unplanting*. Experiments conducted on synthetic data confirm previous observations that the success of the collective depends on its relative size but also highlight the critical role of the collective's absolute size. The results show a small gap between the effective success rate and the theoretical lower bounds proposed. The authors also show that their lower bound compares favorably to Hardt et al. (2023) for signal planting.

**Claims And Evidence:**

The two main claims of the paper are:
1. The ability of the proposed framework to assist in estimating the most effective strategy.
2. The ability to provide lower bounds on success, with parameters controlled by the collective.

Arguments supporting both claims are well presented in the paper and backed by experimental results, except in the case of *signal erasing*, for which no experimental evaluation is provided.

**Essential References Not Discussed:**

I think the authors provide an excellent coverage of the existing works.

**Experimental Designs Or Analyses:**

I have reviewed all the experiments. Please refer to the section "Other Strengths and Weaknesses" for detailed comments on the experiments.

**Methods And Evaluation Criteria:**

The authors use Hoeffding’s inequality to derive the lower bounds and conduct experiments on synthetic data to evaluate their usefulness. They select an appropriate baseline (Hardt et al., 2023) for comparison. Overall, the methods and evaluation approach appear reasonable. The authors also discuss the possibility of employing other concentration inequalities to derive tighter bounds.

**Other Comments Or Suggestions:**

- Line 159. The second probability should be population probability
- Line 174. "Fo each feature" -> "For each feature"
- Line 403: y∗ ∈ {Good, Average, Poor} -> y∗ ∈ {Good (G), Average (A), Poor (P)}

**if possible:**
- Propose an alternative notation for $n_{est}$ as it can lead to confusion with $N_{test}$
- Provide concrete examples illustrating each of the three objectives, similar to those given in Hardt et al. (2023) for signal planting and signal erasing (e.g., content creators). Ideally, introducing a running example early, when the objectives are first defined, would help readers better appreciate the nuances between scenarios.
- Line 114: wrap the assumption on collective access to $N$ under Assumption definition (e.g., **Assumption ($A_0$)**)

**Other Strengths And Weaknesses:**

I found the research question of estimating the success rate of collective action based on pooled data very interesting and original. This paper contributes meaningfully to understanding how collective action can be implemented in practice.

Overall, the paper is very well written. Each objective and assumption is clearly defined, and the corresponding strategies and lower bounds on success are explicitly presented. The authors also provide intuitive explanations of the strategies and appropriately discuss the limitations associated with some key assumptions.

My main concerns relate to the experiments and their realism. The data-generation process is not clearly described; for instance, details about the relationship between features and labels are missing. Additionally, it is unclear why only a single configuration for the parameter $\epsilon$ was chosen.

**Questions For Authors:**

- Did you conduct any experiments for configurations with $\epsilon > 0$? If so, could you describe the general trend observed?
- In practice, what performance did you observe for the lower bound on the success of *signal erasing*?
- Line 590: "Additionally, each car is assigned a Car Evaluation label based on a scoring system that considers various factors such as safety rating, fuel type, warranty length, and others" Could you please provide more details on the scoring system used?
- Were experiments repeated multiple times for each fixed value of $n$ (for example, in Fig. 1)?

**Relation To Broader Scientific Literature:**

This paper contributes broadly to discussions on how affected parties can take a more active role in the implementation of responsible AI. It is closely related to Hardt et al. (2023) and advocates for estimating lower bounds on a strategy's success using information accessible to the collective. Thus, the proposed framework can be viewed as a step toward enhancing the autonomy of collectives.

**Theoretical Claims:**

I reviewed Appendix E to better understand how the lower bounds are derived. However, I might not have fully understood certain aspects of the proofs presented in Appendix E.

---

> ### Author Rebuttal · Authors · 2025-03-28
>
> We would like to thank the reviewer for their detailed and constructive review, and their overall positive feedback. We also appreciate them pointing out the typos. We will update $n_{est}$ to $n_{estim}$​, add assumption $A_0$ regarding the collective's access to $N$, and include a running example if it is considered helpful. We will also make sure to include details about the relationship between features and labels in the synthetic dataset.
>
> **Concerning the synthetic dataset, which has also been mentioned by other reviewers**: our paper provides finite sample guarantees for collective action in classification, and the bounds we derive depend on several parameters. The main purpose of our experiments is to illustrate the bounds we obtained and to gain insights into the influence of the various parameters. We considered it more relevant to use a synthetic dataset to better understand the influence of these parameters. For example, we can deliberately manage the label imbalance (see our response to Reviewer CB4v) and clearly demonstrate that the adaptive strategy outperforms the naive strategies in Figure 3, even outperforming the naive strategy based on the second most frequent global label after $y^*$ (see Figure 3 and Table 1). Furthermore, for a fixed $g$, we can investigate how the composition of the training set elements within the signal set impacts the results using a synthetic dataset, but not with a real-world dataset. In ongoing work we will be working with real-world datasets, but for a first study of this methodology we felt that the priority was to understand its properties rather than simply exhibit an application which would give limited insight into whether our conceptual understanding of the parameters is on solid ground.
>
> *Experiments for configurations with $\epsilon > 0$*
>
> Yes, the lower bounds will become weaker as $\epsilon$ increases.
>
> *Lower bound on the success of signal erasing*
>
> The lower bounds are trivial (equal to zero) because the estimation terms are too large, as they depend on the cardinality of $\mathcal{X}$. We briefly mentioned this in the appendix (lines 1104-1105) but we can also include it in the main text if it is considered helpful.
>
> *Experiments repeated multiple times*
>
> We ran the experiments only once. We expect little variance, except for values very close to the steps, which do not affect the overall trend of the curves highlighted in the paper. However, we can still run additional experiments and average the results if deemed helpful.
>
> We thank the reviewer again for their valuable feedback.

---

### Official Review · Reviewer_CB4v · 2025-03-11

**Overall Recommendation:** 3

**Summary:**

This paper addresses the problem of collective action under finite samples. There are $N$ consumers each with a datapoint $(x,y)\sim \mathcal{D}$ where $x\in \mathcal{X}$ and $y\in \mathcal{Y}$. Of these $N$ consumers $n$ consumers plan to collude, so that after learning from the $N-n$ clean samples and $n$ corrupted samples, the value of a metric $S$ on a test set $D_{test}$ of size $N_{test}$ is large.

Previous work (Hardt et al 2023) established lower bounds on $S$ for $n, N\to \infty$ with the ratio $\alpha = \frac{n}{N}$ being constant. In this paper, the authors try to obtain lower bounds for finite $n$ and $N$. They aim to obtain lower bounds on $S$ that can be computed by the $n$ consumers before collusion, so that they can guarantee success of collusion, i.e., large $S$.



The authors consider $3$ problems i) Signal planting, ii) Signal Unplanting and iii) Signal Erasure. Here, signal planting and erasure had been introduced by (Hardt et al 2023). In each of these problems, there is a signal set $\tilde{\mathcal{X}}$ defined as the range of a  map $g:\mathcal{X} \to \mathcal{X}$. In signal planting, and unplanting, the goal is to add/remove a specific label $y^\star$ for all features in the signal set $\tilde{\mathcal{X}}$. In the signal erasure, the goal is to learn a predictor that has same output on $g(x)$ and $x$, thereby completely eroding the impact of $g$ and thus the signal set.


For each of these problems, the authors have atleast one strategy defined by a specific transformation $h:\mathcal{X}\times \mathcal{Y} \to \mathcal{X}\times\mathcal{Y}$, that transforms the corrupted datapoints. The main theoretical claims of the paper are that a lower bound on the empirical metric $\hat{S}$ for these strategies can be computed efficiently from only the original data of the $n$ corrupted consumers (See Algorithms 1-4). The corresponding finite-sample lower bounds are provided in Theorems 3.3, 3.5, 3.7 and 3.9.


Further, the authors test their signal planting/unplanting and erasure bounds on a large synthetic dataset of cars with categorical features. They show small deviations between the empirical quantities and their bounds. Further, their methods are more accurate than (Hardt et al 2023) in predicting the change in metric.

**References**
- (Ben-Dov et al 2024) The Role of Learning Algorithms in Collective Action. ICML.

**Claims And Evidence:**

Most claims in the submission are well supported. Here are the main ones.

1. The theorems in Section 3 provide finite-sample easy to compute lower bounds allowing on the metric $S$ thus yielding Algorithms 1-4.

2. Dependence of success metric not just on corruption ratio $\frac{n}{N}$ but also the absolute value of $n$: A nice explanation of this is provided in Appendix E.5 and Figure 7 in addition to Figure 2. The authors might consider moving some parts of this section from appendix to the main paper. Note that this was the only experiment related to signal erasure.

3. Experiments on signal planting/unplanting to check the tightness of the lower bounds from theorems were provided in Figures 1 and 3.


4. Better empirical performance than (Hardt et al 2023) in Figure 4 for large number of samples, i.e., population limit. The authors claim that the metric $S$ increases in a staircase-fashion as $\frac{n}{N}$ increases. For this example, this staircase is visible with $3-4$ stairs however, that is not enough evidence to claim existence of such a property. Either more experiments on different datasets/settings should be done or adequate theory provided for such staircase-property before such claims could be made.

5. Absence of experiments on signal erasure task: The paper would have been more complete with these experiments.

**Essential References Not Discussed:**

Most essential references were discussed. A recent reference is (Ben-Dov et al 2024) which the authors might have missed. This reference uses a Distributionally Robust Optimization Approach but with infinite samples like (Hardt et al 2023).

**Experimental Designs Or Analyses:**

Please see the "Methods and Evaluation Criteria" section for all comments.

**Methods And Evaluation Criteria:**

- **Dataset not real** While all experiments are performed on a single large sythetic dataset with each datapoint being a car and its condition (one of $4$ conditions). The feature vector of a car correspond to its different parts, and the condition of the car is calculated as a score of these features. While the experiments on this dataset are complete (except signal erasure task), no real dataset have been used. This is a bit surprising considering several appropriate experimental setups for this problem in (Hardt et al 2023, Ben-Dov et al 2024). Note that I  don't expect the authors to run any additional experiments in the review period, but in case of acceptance, they should add atleast $1$ task on a real dataset.



- **Synthetic Dataset with Label Imbalance**: The other issue with this dataset is that it appears to be synthetic, but it has qualities of a real dataset, for instance label imbalance between "Good" and "Average" labels. If this was truly a synthetic dataset, then the authors should have controlled the label imbalance so that it does not affect the results. Can the authors verify if this dataset is truly synthetic and the authors purposefully put in this imbalance or if some parts of it have been obtained from real datasets which would explain this label imbalance? Note that real datasets are generally harder than synthetic ones. This label imbalance leads to large discrepancies in the $2$ subfigures on the right in Figure 5.



- **Reproducibility**: The authors didn't mention the number of random seeds, the amounnt of compute resources or any variances in their experiments. While these are not extremely important to the main result, they are important for reproducibility. I would recommend including this in the final version in case of acceptance.

**Other Comments Or Suggestions:**

- **Optimal value of $n_{est}$**: In signal unplanting, the authors provide experiments on there being an optimal value of $n_{est}$ in Figure 3(a). It seems like the authors can provide theoretically optimal values of $n_{est}$ as well, atleast orderwise.


- **Typos**:
    1. Line 173, 2nd column : "Fo" -> "For".
    2. Line 876 : "than" -> "as".
    3. Lines 913-914 : two "conditionnally" -> "conditionally".

**Other Strengths And Weaknesses:**

## Other Weakness
- **Convex action (Sections 4.1 \& 4.2, Hardt et al 2023)**: The authors did not extend their finite-sample guarantees to the case of collusion when the goal is to minimize a convex empirical risk as described in (Hardt et al 2023). This is an extremely important application of these frameworks. Could the authors comment on whether their results could be directly applied in these cases? Would it be possible if the model classes have strong uniform generalization bounds? If applying their scheme is hard, then they can just point out the main difficulties in this setup.

**Questions For Authors:**

- Line 428, 1st column : "This mirrors the shape of curves observed in signal unplanting in Figure 3, which would also appear in Figure 1 and Figure 2". Unfortunately, I am not able to identify any staircases in Figures 1-3 and all of them look more like noisy sigmoids.

- Does the bound on $\eta< \frac{1}{\\# \tilde{\mathcal{X}}}$ result in additional constraints in Theorem 3.9? Also, in Lines 1105-1106, can the poor value of the bound in Theorem 3.9 be possibly improved by slightly changing the definition of $\eta$ to a quantity in expecatation, or is it unavoidable?

- What was the issue with the earlier proof of (Hardt et al 2023) for $\epsilon>0$ (Appendix F.4)? Does Definition F.3 which is the current version of (Hardt et al 2023) address it ?

**Relation To Broader Scientific Literature:**

This paper provides an efficient way for consumers to determine their success metric if they collude. This is a strict improvement over the known collusion results of (Hardt et al 2023). Their related works section is thorough. Their discussion section also clearly connects their work and its limitations to other related problems.

**Theoretical Claims:**

- **Correctness** : The proofs seem correct to me. The core idea that the authors use is this, first using Hoeffding's inequality, they bound the difference between population and finite-sample terms. Then, they take an appropriate union bounds over all these concentration bounds so that they hold simultaneously. After this, all they need is to plug in the expression of the metric $S$ and use the given strategy and the concentration inequalities, until they end up with terms only dependent on datapoints on $n$ consumers.

- **Justification of Assumptions 1 and 2**: From what I understand in (Hardt et al 2023), these assumptions are not explicitly used. The value $\tau$ in (Hardt et al 2023) seems similar to $\eta$ in Assumption 1, although it is required for every feature in $\tilde{\mathcal{X}}$, while $\tau$ is defined only on expectation wrt the true distribution. I understand the authors' justification for these assumptions, but can they compare this with corresponding assumptions, or lack of it, in comparable works like (Hardt et al 2023).

---

> ### Author Rebuttal · Authors · 2025-03-28
>
> We would like to thank the reviewer for their detailed and technical review, which allows for a discussion of some of the paper’s theoretical aspects. We also appreciate their positive feedback and will now address the questions raised.
>
> *Better empirical performance than (Hardt et al 2023) in Figure 4 for large number of samples*
>
> Our bounds are not only empirically superior but also theoretically better (see Proposition F.2, for example).
>
> *Either more experiments on different datasets/settings should be done or adequate theory provided for such staircase-property before such claims could be made*
>
> For a finite signal set, we can theoretically prove that our bounds exhibit a staircase shape, unlike Hardt et al., just by using law of total probability and conditioning on the possible values for $g(x)$. We can include a rigorous proof in the paper if it is considered helpful.
>
> *Absence of experiments on signal erasure task / dataset not real*
>
> We refer to our response to Reviewer HD8o.
>
> *Synthetic Dataset with Label Imbalance*
>
> The dataset is truly synthetic and we intentionally introduced this imbalance.
>
> *Reproducibility*
>
> We fixed the random seed for our experiments (see src file).
>
> *Justification of Assumptions 1 and 2*
>
> The parameters $\eta$ and $\tau$ are very different. While $\eta$ relates to the nature of the base distribution and ensures that the most frequent label is sufficiently separated from the second most frequent one, $\tau$ reflects how similar the probability of a label given $x$ is to its probability given $g(x)$. In other words, $\tau$ captures the extent to which $g$ already influences the platform’s decisions, whereas $\eta$ is a simple assumption about the base distribution, entirely independent of $g$ or any other parameters specific to the collective. Regarding Assumption 2, we refer to our response to Reviewer 6HMD.
>
> *Essential References Not Discussed*
>
> Indeed, the paper cited by the reviewer is relevant, and we will ensure it is included in the references.
>
> *Convex action*
>
> As mentioned by the reviewer, convex risk minimization is also an important framework. We believe that gradient-based learning, as outlined by Hardt et al., is also an important framework. While the focus of our paper is only on classification, exploring these setups is an interesting direction for future work. We believe that our setting could be extended to this case even without assuming strong uniform generalization bounds: one can simply estimate gradients directly using samples from the base distribution and the gradient-neutralizing distribution. One additional difficulty is to compute a gradient-neutralizing strategy. Within our setting, the collective does not directly have access to expected gradients but rather to estimations and the collective may only be able to compute an approximate gradient-neutralizing strategy. This would yield additional error terms.
>
> *Identify staircases in Figures 1-3*
>
> The steps in the staircase are closely spaced, and the bounds are computed only for fixed values of $n$; this is why the staircase resembles a sigmoid curve.
>
> *Bound on $\eta$*
>
> We do not assume that $\eta < \frac{1}{Card(\tilde{\mathcal{X}})}$. Regarding the second part of the reviewer’s remark: it is possible to redefine $\eta$ to discard features not in $\tilde{\mathcal{X}}_0 \subseteq \tilde{\mathcal{X}}$, where $\tilde{\mathcal{X}}_0$ is a subset of $\tilde{\mathcal{X}}$ such that $\tilde{x} \sim \tilde{\mathcal{D}}$ has a high probability of belonging to $\tilde{\mathcal{X}}_0$. This aligns with the reviewer’s intuition of asking whether a similar definition of $\eta$ in expectation is possible. In this case, we can derive theoretical guarantees by conditioning the success and saying that the success is equal to the same probability conditional on the fact that the feature $g(x)$ is in $\tilde{\mathcal{X}}_0$ times the probability that $g(x)$ is in $\tilde{\mathcal{X}}_0$, plus a term that depends on the probability of $g(x)$ not being in $\tilde{\mathcal{X}}_0$, that we can typically discard.
>
> *Issue with the earlier proof of (Hardt et al., 2023)*
>
> There was a mistake in algebraic manipulations leading to an erroneous bound. In the current version of Hardt et al., Definition F.3 is used, but the bounds remain suboptimal due to unnecessary inequalities.
>
> We sincerely thank the reviewer for the thorough review and for pointing out typos.

---

> > ### Comment · Reviewer_CB4v · 2025-04-01
> >
> > Thank you for the detailed response. My questions have been answered, and I do not have any further questions.
> >
> > Just one remark. For bound on $\eta$, take Assumption A1 and sum it over all $x\tilde{\mathcal{X}}$. The LHS is still a probability so it can be bounded by $1$ and the RHS can be lower bounded by $\eta |\mathcal{X}|$. This is a consequence of Assumption A1.

---

### Official Review · Reviewer_6HMD · 2025-03-17

**Overall Recommendation:** 3

**Summary:**

The paper explores the statistical inference for collective action in learning platforms, in particular, the paper examines not only the classic signal planting procedure introduced by Hardt et al, but also signal unplanting as well as signal erasing, both of which require statistical inferences for defining the optimal strategy $h$. In the end, the authors develop a framework that provides a theoretical and algorithmic treatment and preform empirical studies on synthetic datasets.

**Claims And Evidence:**

The claims are supported by clear and convincing mathematical arguments.

**Essential References Not Discussed:**

NA

**Experimental Designs Or Analyses:**

I did check the experimental design and they are sound and valid.

**Methods And Evaluation Criteria:**

The proposed method and evaluation criteria make sense for the problem.

**Other Comments Or Suggestions:**

It would be nice if the authors could add some intuitions behind theorem 3.7, especially how it differs from the previous theoretical guarantee (Theorem 3.3).

**Other Strengths And Weaknesses:**

Overall, the paper is well-written and well-explained. The proposed signal planting and signal erasing methods are intuitive. The experimental section is a bit weaker since it's a simulated setting. Another weakness is that some theoretical claims rely on particular assumptions or particular signal strategies, for example, it's unclear to me whether theorem 3.7 would still hold if the signal unplanting strategy is different from equation (3), and whether Theorem 3.9 would still hold if the transformation g is not idempotent. Some discussions on how violation of these assumptions/particular strategy could change/affect the theoretical result could be helpful.

**Questions For Authors:**

1) How should I understand the inequality in section 3.3.1?

2) Does Theorem 3.7 still hold if we randomly choose a label other than $y^*$ instead of $\hat{y}_{\tilde{x}}$ according to equation (3)?

3) In section 3.4 (signal erasing), it seems like the success of the proposed signal erasing procedure relies on the transformation function g to be idempotent, namely $g(g(x)) = g(x)$. What are some examples of idempotent vs. non-idempotent transformations? How practical is it in real life?

**Relation To Broader Scientific Literature:**

The key contribution of the paper is on relating statistical inference to the literature of collective action, which is missing in the previous work in the literature.

**Theoretical Claims:**

I checked the soundness of the theoretical claims.

---

> ### Author Rebuttal · Authors · 2025-03-28
>
> First, we would like to thank the reviewer for their time and their insightful comments. As mentioned, our main contribution is introducing a statistical inference component to the collective action framework, which is crucial in practice, as collectives often seek guarantees about the effectiveness of their strategies. We also appreciate the reviewer’s thoughtful questions and will ensure additional clarifications in the final version to enhance intuition and understanding.
>
> *It would be nice if the authors could add some intuitions behind theorem 3.7, especially how it differs from the previous theoretical guarantee (Theorem 3.3).*
>
> We agree that a discussion here would indeed help provide the reader with more intuition, particularly in comparison to signal planting (Theorem 3.3). In signal planting, the strategy is simple: flood the platform with label $y^*$. In signal unplanting, an intuitive approach is to change the label by choosing the most probable label $\hat{y}\_\tilde{x}$ after $y^∗$ for every $\tilde{x}$, which is exactly the meaning of equation (3). However, $\hat{y}\_\tilde{x}$ is unknown, so we propose estimating it using a subset of the collective of size $n_{est}​$. This results in a two-step inference method, where we first adaptively estimate the label $\hat{y}\_\tilde{x}$, and then use concentration inequalities that depend on $\hat{y}\_\tilde{x}$. To ensure independence and correctly apply Hoeffding’s inequality, we carefully split the datasets of size $n_{est}​$ and the rest of the collective of size $n - n_{est}$. This is why the estimation term for $\Delta_{\tilde{x}}$ in Theorem 3.7 is $R(n - n_{est})$, rather than $R(n)$ in Theorem 3.3. This directly quantifies the impact of not knowing the optimal label to play by the collective (contrary to signal planting where the optimal label to play is known): the counteracting term in the bound is more important since $R(n - n_{est}) > R(n)$.
>
> *1.How should I understand the inequality in section 3.3.1?* and *2. Does Theorem 3.7 still hold if we randomly choose a label other than $y^\*$ instead of $\hat{y}\_\tilde{x}$ according to equation (3)?*
>
> To build intuition, let’s consider the following simplified case: the signal set $\tilde{\mathcal{X}}$ consists of a single feature, and the label frequencies associated with the training set without the collective are as follows: 100 for $y_0=:y^*$, 70 for $y_1$​, 60 for $y_2$​, and 50 for $y_3$​. In this case, the optimal strategy seems to be estimating the most probable label $\hat{y}$ after $y_0$ (here,$\hat{y} = y_1$​) and flooding the platform with it. However, $\hat{y}$​ is unknown by the collective. The collective could naively flood the platform with either $y_2$​ or $y_3​$ as well. This corresponds to planting the signal with those labels, which is exactly the meaning of the inequality in Section 3.3.1. A more effective approach is to estimate $\hat{y}​$, which we formalize in the paper.
> Regarding the suggestion of using randomness: while it is possible, we believe it is suboptimal, even compared to naive strategies, which is why we did not study it. To get convinced, assume a collective of size $n=60$. Naively flooding the platform with $y_i$ ($i \in \{1,2,3\}$)​ succeeds since $y_i$​ becomes the most frequent label for $\tilde{x}$. However, choosing a label randomly among those $\neq y^*$ leads to expected frequencies: $y_0: 100$, $y_1: \approx 90 < 100$, $y_2: \approx 80 < 100$, and $y_3: \approx 70 < 100$, resulting in an unsuccessful collective action.
>
> *3. In section 3.4 (signal erasing), it seems like the success of the proposed signal erasing procedure relies on the transformation function $g$ to be idempotent. What are some examples of idempotent vs. non-idempotent transformations? How practical is it in real life?*
>
> We provide examples of idempotent functions $g$ in lines 311-319. This assumption is generally not restrictive, especially in tabular data or opaque watermarking.
>
> **Concerning weaknesses:**
>
> *The experimental section is a bit weaker since it's a simulated setting*
>
> We refer to our response to Reviewer HD8o.
>
> *Another weakness is that some theoretical claims rely on particular assumptions or particular signal strategies*
>
> We study strategies that are intuitively optimal and provide theoretical guarantees. Establishing a unifying framework for analyzing strategy optimality may be an interesting direction for future work. The guarantees we derive depend on assumptions, such as the idempotency of $g$ for signal erasing (line 1025). We do not claim that this assumption is necessary, but we have not found a way to derive guarantees without it. If helpful, we can clarify why we use this assumption, in contrast to Hardt et al.
>
> We sincerely thank the reviewer for their valuable feedback, which helps improve the paper, and we hope we have addressed their concerns.

---

### Decision · Program_Chairs · 2025-05-01

**Decision:**

Accept (oral)

**Comment:**

The authors provide a novel theoretical characterization of the effectiveness of collective action in machine learning systems. In particular, they extend on the framework of Hardt et al 2023 to account for how the collective’s actions could be accounted for, and possibly mitigated, by the firm. This leads to new concepts for the literature on collective action, such as signal unplanting and signal erasing. While the empirical validation using synthetic data was identified as a potential limitation, the reviewers praised the intuitive theoretical analysis, which sheds light on this emerging area.